# Pbp1, the yeast ortholog of human Ataxin-2, functions in the cell growth on non-fermentable carbon sources

Dang Thi Tuong Vi[1], Shiori Fujii[1], Arvin Lapiz Valderrama[1,2], Ayaka Ito[1], Eri Matsuura[1], Ayaka Nishihata[1], Kaoru Irie[1], Yasuyuki Suda[1,3], Tomoaki Mizuno[1], Kenji Irie[1,2]*

1 Department of Molecular Cell Biology, Graduate School of Comprehensive Human Sciences and Faculty of Medicine, University of Tsukuba, Tsukuba, Japan, 2 Program in Human Biology, School of Integrative and Global Majors, University of Tsukuba, Tsukuba, Japan, 3 Live Cell Super-Resolution Imaging Research Team, RIKEN Center for Advanced Photonics, Wako, Saitama, Japan

* kirie@md.tsukuba.ac.jp

**Data Availability Statement:** All relevant data are within the paper and its Supporting Information files.

## Abstract

Pbp1, the yeast ortholog of human Ataxin-2, was originally isolated as a poly(A) binding protein (Pab1)-binding protein. Pbp1 regulates the Pan2-Pan3 deadenylase complex, thereby modulating the mRNA stability and translation efficiency. However, the physiological significance of Pbp1 remains unclear since a yeast strain harboring *PBP1* deletion grows similarly to wild-type strain on normal glucose-containing medium. In this study, we found that Pbp1 has a role in cell growth on the medium containing non-fermentable carbon sources. While the *pbp1Δ* mutant showed a similar growth compared to the wild-type cell on a normal glucose-containing medium, the *pbp1Δ* mutant showed a slower growth on the medium containing glycerol and lactate. Microarray analyses revealed that expressions of the genes involved in gluconeogenesis, such as *PCK1* and *FBP1*, and of the genes involved in mitochondrial function, such as *COX10* and *COX11*, were decreased in the *pbp1Δ* mutant. Pbp1 regulated the expressions of *PCK1* and *FBP1* via their promoters, while the expressions of *COX10* and *COX11* were regulated by Pbp1, not through their promoters. The decreased expressions of *COX10* and *COX11* in the *pbp1Δ* mutant were recovered by the loss of Dcp1 decapping enzyme or Xrn1 5'-3'exonuclease. Our results suggest that Pbp1 regulates the expressions of the genes involved in gluconeogenesis and mitochondrial function through multiple mechanisms.

## Introduction

Gene expression consists of several steps involving the whole flow of information from DNA to protein through mRNA [1]. Regulation at the transcriptional level is of the highest importance as an early control will prevent the synthesis of unnecessary intermediates, saving energy and materials. Post-transcriptional regulation provides an additional control layer to respond rapidly to changes in the environment [2]. Post-transcriptional regulation begins with the control of transcription-coupled mRNA maturation procedures including the addition of the cap 7-methylguanosine to the 5' end, splicing to remove introns, and the addition of poly(A) tail to

**Funding:** This research was supported by JSPS KAKENHI Grant Number 18K06053 (to KI).

**Competing interests:** The authors declare that they have no conflict of interest.

the 3' end [3]. Once exported to the cytoplasm, mRNA can be either translated into polypeptides or degraded. These two competing processes are central in the post-transcriptional regulation since they determine the rate of protein synthesis and set the steady-state level of mRNA of a specific gene. In these processes, the 5' cap, 3' poly(A) tail, and 3' untranslated region (UTR) of mRNAs are vital because they contain *cis*-acting elements that facilitate the interaction of mRNA with RNA-binding proteins responsible for translation initiation or mRNA degradation [4].

The growth of yeast cells relies on various compounds that serve as energy sources for the synthesis of essential metabolites [5]. In yeast, there is a hierarchy of carbon source utilization wherein glucose or fructose is favored over saccharides such as raffinose [6]. Even in the presence of oxygen, yeast cells predominantly perform aerobic fermentation [7]. Non-fermentable carbon-sources, such as glycerol, ethanol, acetate, and lactate, are the least preferred as their catabolism needs oxidative phosphorylation. The yeast cells utilize transcriptional and post-transcriptional regulations to fine-tune gene expression and efficiently respond to the changes in the nutrient source. In glucose-rich media, the genes required for cell growth and proliferation are activated, while the genes associated with stress response or involved in processing alternative carbon sources are repressed. Previous studies revealed a massive change in transcriptome content when cells are released from glucose deprivation or transferred from the rich medium to ethanol [8–11]. In these reports, microarray analysis showed that the genes coding for key enzymes functioning in processes related to energy metabolism as well as critical proteins required for transcription and translation are enriched in those exhibiting a glucose-dependent expression pattern. Glucose signals can also modulate gene expression by changing the fate of mRNAs in the cytoplasm. A previous study showed that glucose depletion rapidly inhibits translation initiation in yeast [12]. This regulation appears to be selective since the re-addition of glucose into the media can re-activate only a sub-population of mRNAs that were translationally repressed [13]. Post-transcriptional control also operates individually through messenger ribonucleoprotein (mRNP) complexes such as the regulation by Puf-family RNA-binding proteins which bind to 3' UTR of target mRNAs [14–16]. For example, Puf3 binds to mRNAs coding for proteins related to mitochondrial function. When glucose is depleted, Puf3 becomes heavily phosphorylated and promotes the translation of its bound transcripts [17]. However, little is known about how the signals from the changes in glucose concentration are transduced to RNA-binding proteins.

Pbp1 is a yeast protein that binds to Pab1 (Poly(A)-binding protein) [18]. The previous report showed that Pbp1 operates in both stressed and non-stressed conditions [18]. Pbp1 participates in mRNA processing and turnover by promoting polyadenylation [19, 20]. A model in which Pbp1 prohibits the cleavage of poly(A) tail executed by the poly(A) nuclease (PAN) is suggested by the observation that deletion of the *PBP1* gene led to accelerated poly(A) tail trimming [19, 20]. Pbp1 can also affect the protein level without changing the mRNA level by forming a complex with Mkt1 to positively regulate the translation of *HO* mRNA [15]. During glucose deprivation, Pbp1 localizes to stress granules, stress-induced cytoplasmic foci containing temporarily non-translated mRNAs [21, 22], suggesting its role in nutrient-responsive regulation of mRNA [22]. Furthermore, the deletion of *PBP1* significantly reduces the number of stress granules upon glucose deprivation but not upon sodium azide treatment, suggesting stress-specific functions of Pbp1 in stress granule assembly [21]. Following glucose withdrawal, Pbp1 is directly phosphorylated by Psk1 kinase [23]. This phosphorylated and activated form of Pbp1 sequesters target of rapamycin complex (TORC) 1 into stress granules and inhibits its activity [24]. Recently, it is demonstrated that a methionine-rich region is required for Pbp1 self-association, which is fundamental for the sequestration and suppression of TORC1 during respiration growth [25, 26].

In this study, we investigated the role of Pbp1 on cell growth in a medium containing a non-fermentable carbon source. While the *pbp1Δ* mutant showed a similar growth compared to the wild-type cell on media containing glucose as a carbon source, the *pbp1Δ* mutant showed a slower growth on the medium containing glycerol and lactate. Our analyses have revealed that Pbp1 regulates expressions of specific genes involved in gluconeogenesis and mitochondrial functions.

## Materials and methods

### Strains and media

*Escherichia coli* DH5α strain was used for DNA manipulations. The yeast strains and plasmids used in this study are isogenic derivatives of the W303 strain and listed in S1 Table. Gene deletions were conducted to replace the target gene with resistance cassettes by homologous recombination using standard PCR-based method [27]. Colony PCR was conducted with forming clones to confirm complete deletion at the expected locus. The media used in this study included YPD (1% yeast extract, 2% peptone, and 2% glucose), YPGL (1% yeast extract, 2% peptone, 3% glycerol, and 2.8% lactate), YPRaff (1% yeast extract, 2% peptone, and 2% raffinose) and synthetic complete (SC) medium (0.67% yeast nitrogen base without amino acids (BD Difco, NJ, USA), 0.5% casamino acid (BD Bacto, NJ, USA), 0.01% adenine, 0.01% uracil, and 2% glucose. SC media lacking amino acids or other nutrients (e.g. -uraGlu corresponding to SC lacking uracil, -uraGL corresponding to SC lacking uracil and containing 3% glycerol, and 2.8% lactate) were used to select the transformants. Solid media (YPD, YPGL, and YPRaff) were prepared using 2% (w/v) agar (Rikaken, Aichi, Japan). General procedures were performed as described previously [28].

### Plasmids

Plasmids used in this study are listed in S2 Table. pCgLEU2, pCgHIS3, and pCgTRP1 are pUC19 carrying the *Candida glabrata LEU2*, *HIS3*, and *TRP1* genes, respectively [29]. YCpPBP1 is YCplac33 carrying the *PBP1* gene. Others were constructed as described in the following sections.

### Cell collection, RNA extraction, microarray data and qRT-PCR

Cells from overnight cultures were inoculated into 20 mL fresh YPD to 0.5 OD/mL. After 4 hours cultured in YPD, 10 mL of media were transferred into a centrifuge tube. The cells were then spun down, washed, and re-suspended in 10 mL of YPGL. Cells were collected at 4 hours after cultured in YPD and YPGL. Approximately 6 OD of cells were collected, and total mRNAs were then prepared using ISOGEN reagent (Nippon Gene, Toyama, Japan) and the RNeasy Mini kit (Qiagen, Dusseldorf, Germany). First strands of cDNAs were generated using the Prime Script RT reagent Kit (Takara, Shiga, Japan). The cDNAs were quantified by a quantitative real-time RT-PCR (qRT-PCR) method using either 7500 fast or QuantStudio 5 real-time RT-PCR systems (Applied Biosystems, MA, USA) with SYBR Premix Ex Taq (Takara, Shiga, Japan). The fold changes in mRNA levels were calculated by using the $2^{-\Delta\Delta Ct}$ method and normalized against the *ACT1* or *SCR1* reference gene. The primers for RT-PCR are listed in S4 Table. The microarray analysis was performed by the KURABO Bio-Medical Department (Osaka, Japan) using the Affymetrix GeneChip Yeast Genome 2.0 Array (Affymetrix, Santa Clara, CA, USA). Microarray data sets are available at the Gene Expression Omnibus at http://www.ncbi.nlm.nih.gov/geo (GEO accession number GSE124908).

## Construction of GFP reporter gene

The YCplac33-*pPCK1-GFP* plasmid was constructed as follows. The fragment encoding *PCK1* promoter was obtained by PCR from genomic DNA using the primer pair CAAGCTTGCATG CCTGCAGGTCGACCACATGTCGACGAGTTTGTC (tv1*PCK1*pro-33-F) and GTTAATTAACCC GGGGATCCGGGACATGTTGTTATTTTATTATGG (tv2*PCK1*pro-GFP-R). The fragment encoding GFP reporter gene followed by *ADH1* terminator were amplified from pFA6A-GFP (S65T)-KanMX6 plasmid (Addgene plasmid # 39292) using the primer pair CCA TAATAAAA TAACAACATGTCCCGGATCCCCGGGTTAATTAAC (tv3*GFP-PCK1*pro-F) and TTGTAAAAC GACGGCCAGTGAATTCAGATCTATATTACCCTGTTATCC (tv4*ADH1*-33-R). Two fragments were inserted into between *Xba*I and *Eco*RI sites of the YCplac33 plasmid using gap repair cloning [30]. The plasmids harboring *GFP* reporter gene for the other genes were constructed by the same method using pairs of primers listed in S3 Table. Primer tv4*ADH1*-33-R was used in the construction of *GFP* reporter for all other genes. All the plasmids used in this study are listed in S2 Table.

## 3' UTR exchange construction

To change 3' UTR of endogenous *COX10* gene to *ADH1* terminator, a fragment coding for HA tag with *ADH1* terminator and kanamycin resistance selection marker was amplified from pFA6a-3HA-kanMX6 plasmid (Addgene plasmid # 39295) using the primer pair GGATTGGA TATATCCTGGTGAAGCAAAGCGACCACAGGAACGATTTACGGATCCCCGGGTTAATTAA and GACTGCCCTTTAAGCGTTGTCTCTTTATCTCATTGTACTAATGGAATTCGAGCTCGTTTAA AC, introduced into the *PBP1*/*pbp1Δ* strain (diploid strain heterozygous for *PBP1* deletion). The diploid cells were sporulated and the spores were dissected to obtain wild-type and *pbp1Δ* strains that harbor the *COX10* gene with *ADH1* terminator. To change *ADH1* terminator of *GFP* gene driven by *COX10* promoter to *COX10* 3' UTR, a fragment containing *COX10* promoter followed by *GFP* gene was amplified from YCplac33-*pCOX10-GFP* (S2 Table) using the primer pair consisting of tv5COX10pro-33-F and GTCTCTTTATCTCATTGTACTAATGTT ATTTAGAAGTGGCGCGCGCCCTATT (tvc54COX10-R). *COX10* 3'UTR (500 bp downstream of the stop codon) were amplified from genomic DNA with the primer pair AATAGGGCGC CACTTCTAAATAACATTAGTACAATGAGATAAAGAGAC and TTGTAAAACGACGGCCAGT GAATTCCTTGACAGCGAAAGATATAGCTAAG. Two fragments were inserted into between *Xba*I and *Eco*RI sites of the YCplac33 plasmid using gap repair cloning [30]. A similar construction for *COX11* gene was constructed by the same method using pairs of primers listed in S3 Table.

## Western blot analysis

Yeast cells were cultivated first overnight (16-hours), then a solution having $OD_{600} = 0.5$ (optical density measured at a wavelength of 600 nm) was prepared. For every sampling time, 10 OD of cells was collected from the cultured liquid media to be used for protein extraction. The cells ($OD_{600} = 10$) of the collected cells were treated with sodium hydroxide for protein extraction [31]. Protein samples were loaded onto an 8% SDS-PAGE gel for protein electrophoresis and then transferred to a PDVF membrane (Merck Millipore, Molsheim, France) for Western blot analysis. The band detection was carried out by using a LAS-4000 (Fuji Film, Tokyo, Japan) with Immobilon Western (Merck Millipore, Molsheim, France), while the signal intensities were quantified through Image Quant (GE Healthcare, Chicago, USA).

# Results

## Pbp1 positively regulates cell proliferation in non-fermentable carbon source media

After the depletion of glucose and other fermentable carbon sources, yeast will utilize other carbons including non-fermentable compounds. This shift to non-fermentable carbon sources results in the reprogramming of gene expression involved in gluconeogenesis, the glyoxylate cycle, and the tricarboxylic acid cycle [32]. To get an insight into the importance of Pbp1 in response to nutritional cues, we examined the growth of *PBP1* deletion strain (*pbp1Δ*) on three different solid media (Fig 1A). We used raffinose and glycerol plus lactate to simulate the conditions in a less preferable sugar and a non-fermentable carbon source, respectively. The *pbp1Δ* mutant grew similarly to wild-type cells on YPD medium containing glucose as a carbon source. However, the *pbp1Δ* mutant showed slower growth than wild-type cells on YPRaff medium containing raffinose, and much slower growth on YPGL medium containing glycerol and lactate (Fig 1A). We confirm that the *pbp1Δ* mutant showed slower growth than wild-type cells on YPGL medium containing glycerol and lactate by spot assay (Fig 1B) and that the slow growth of the *pbp1Δ* mutant was efficiently complemented by the *PBP1* plasmid (Fig 1C). We also examined the growth of the *pbp1Δ* mutant in liquid media. While the *pbp1Δ* mutant grew similarly to wild-type cells in YPD medium, the *pbp1Δ* mutant showed slightly slower growth than wild-type cells in YPRaff medium and much slower growth in YPGL medium (Fig 1D). These observations are consistent with a previous report showing that *pbp1Δ* mutant strain exhibited reduced competitive fitness in a glycerol medium [33]. Similarly, it has been reported that the *pbp1Δ* mutant showed a decreased growth rate in the media containing glycerol and lactate in a systematic functional screen [34]. All these results suggest that Pbp1 positively regulates cell growth on non-fermentable carbon sources.

## Pbp1 regulates specific genes related to gluconeogenesis and mitochondria function

The growth defect in YPGL medium observed in *pbp1Δ* mutant suggests that Pbp1 regulates the expression of genes that are important for cell growth in non-fermentable carbon sources. To identify potential targets of Pbp1, we compared the gene expression profile between wild-type and *pbp1Δ* mutant strains growing in YPD and YPGL media by microarray analysis. We screened the genes that exhibited a noticeable change in expression (fold change ≥ 2) upon *PBP1* deletion. In YPD medium, 29 genes had a notable (at least two-fold) increase in mRNA level while 9 genes had a remarkable decrease (at least two-fold) in mRNA level in the *pbp1Δ* mutant strain. In YPGL medium, *pbp1Δ* mutant showed an increase in the mRNA levels of 43 genes and a decrease in the mRNA levels of 24 genes (Table 1). Interestingly, among down-regulated 24 genes in the *pbp1Δ* mutant cultured in YPGL medium, 4 genes involved in the gluconeogenesis pathway (*PCK1*, *FBP1*, *ICL1*, and *MLS1*) and 9 genes related to mitochondrial function (*COX10*, *MRPL4*, *MRPS35*, *MSY1*, *DIA4*, *AIM33*, *IBA57*, *GCV2*, and *SLM5*), which were highlighted in underline (Table 1). As these genes function in carbon metabolism and respiration, which are important for yeast cells to live in non-fermentable carbon source medium, they might be critical targets of Pbp1 in cells cultured in YPGL. As applying a cut off value of two-fold change may result in a loss of information, we looked for other genes which have similar functions. In the next step, we combined the information of their expression from the microarray data with those of the underlined 13 genes in Table 1 to obtain Table 2. In this table, the genes that are down-regulated in *pbp1Δ* more than two-fold are highlighted in underline, and the genes that are down-regulated in *pbp1Δ* less than two-fold are also shown (Table 2). These genes were classified into seven groups based on gene ontology (Table 3).

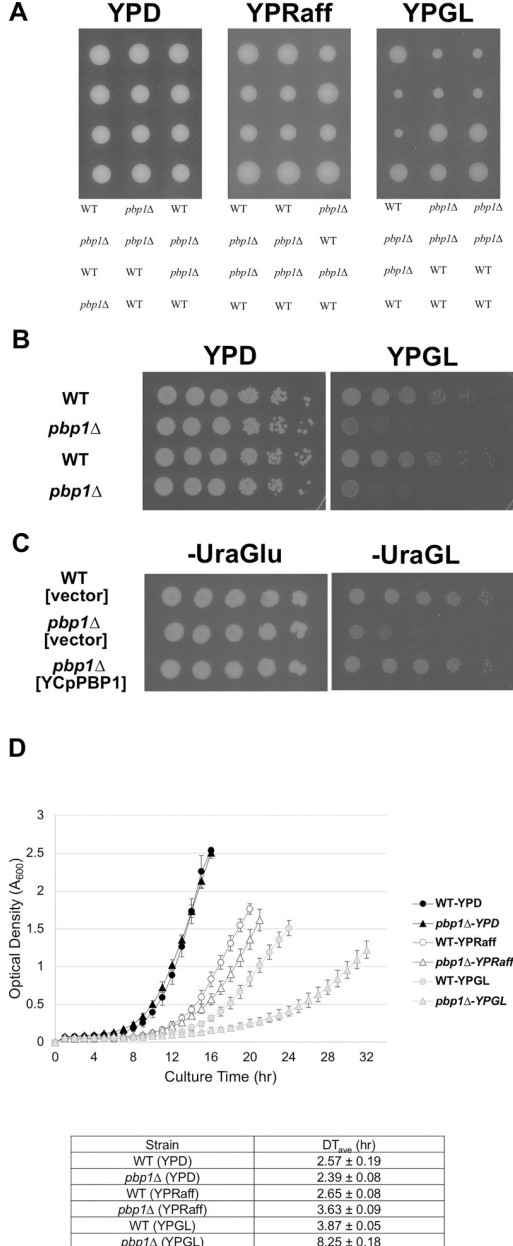

**Fig 1. The growth of the *pbp1Δ* mutant in YPD, YPRaff and YPGL media.** (A) Tetrad analysis. The strains that were heterozygous for *PBP1* alleles were sporulated, and tetrads were dissected onto YPD, YPRaff, and YPGL media. The growth of segregants after 3 days (YPD, YPRaff) and 8 days (YPGL) at 30˚C is shown. Genotypes are indicated on the right sides. More than 50 tetrads were dissected, and representative data are shown. (B) WT and the *pbp1Δ* mutant strains were grown in YPD at 30˚C to the mid log phase. The same optical densities of cells were spotted onto YPD or YPGL plates and then incubated for 2 days (YPD) or 6 days (YPGL) at 30˚C. (C) WT and the *pbp1Δ* mutant harboring the vector or YCpPBP1 were grown in -UraGlu at 30˚C to the mid log phase. The same optical densities of cells were spotted onto -UraGlu or -UraGL plates and then incubated for 3 days (-UraGlu) or 7 days (-UraGL) at 30˚C. (D) The growth curves of WT and the *pbp1Δ* mutant in YPD, YPRaff, and YPGL media. The strains were pre-cultured overnight in YPD and then transferred into fresh YPD, YPRaff, and YPGL to grow at 30˚C. We measured the rate of transmitted light (950 nm) volume at 60-min intervals using an ODBox-C/ODMonitor system (Taitec Corp., Saitama, Japan). The transmission rate was converted into optical density (OD), and the average OD over 60 min was calculated for each sample. The data of doubling time show mean ± SEM (n = 3).

**Table 1. Summary of microarray data: Genes that exhibit a change in mRNA level at least 2-fold in *pbp1Δ* mutant strain comparing with that in wild-type strain in the two media YPD and YPGL.** Names of genes that showed a decreased expression in *pbp1Δ* in YPGL and have functions related to gluconeogenesis or mitochondria are highlighted in underline.

| Media | YPD | | YPGL | |
|---|---|---|---|---|
| Expression change in *pbp1Δ* compared to wild-type | Increase >= two-fold | Decrease >= two-fold | Increase >= two-fold | Decrease >= two-fold |
| Number of gene | 29 | 9 | 43 | 24 |
| Gene names | *MEK1*, *PDC6*, *LOH1*, *PAU20*, *VAM10*, *POT1*, *SRX1*, *BSC4*, *STE3*, *DAK2*, *PAI3*, *HOP2*, *SPR1*, *AMA1*, *IRC18*, *SPS2*, *COS12*, *ATG16*, *SPS22*, *ECM23*, *SPS100*, *RRT15*, *HTX14*, *DIT2*, *SAE3*, *HTX13*, *HTX17*, *FMT1*, *GAT3* | *UTR5*, *YPS5*, *FRE5*, *HTX13*, *ERR1*, *MND1*, *HMS1*, *SRL4*, *NDE1* | *FMP48*, *PUG1*, *YEH1*, *PAU20*, *TIR2*, *SPG4*, *BAG7*, *BNA2*, *CPA1*, *ARG5*, *MAM1*, *SPS19*, *IZH4*, *PCL5*, *SSP1*, *MFα1*, *SPI1*, *IMD1*, *ASN1*, *BNA5*, *ADH5*, *ECM8*, *SUC2*, *SPO11*, *CRR1*, *THI4*, *SPR3*, *STE3*, *HPF1*, *RGI2*, *ARG1*, *IMD2*, *HSP32*, *DSF1*, *GAT3*, *SAE3*, *ARG3*, *AIF1*, *PFS1*, *RRT5*, *DTR1*, *ECM23*, *BSC4* | *PTR2*, *HMS1*, *DAK2*, *TPO2*, <u>*MLS1*</u>, <u>*COX10*</u>, <u>*PCK1*</u>, *SST2*, <u>*MRPL4*</u>, *DAN1*, <u>*FBP1*</u>, *BAR1*, <u>*ICL1*</u>, <u>*MSY1*</u>, <u>*DIA4*</u>, *ARO10*, <u>*AIM33*</u>, <u>*MRPS35*</u>, <u>*IBA57*</u>, <u>*GCV2*</u>, <u>*SLM5*</u>, *SSA3*, *PRM5*, *SRT1* |

We verified microarray data expression level of the genes selected from the Table 2 by conducting qRT-PCR, and summarized the data in Table 3. Deletion of the *PBP1* genes led to the reduction in the mRNA levels of three key enzymes involved in gluconeogenesis and the glyoxylate cycle: in *pbp1Δ* mutant cultivated in YPGL medium, the mRNA levels of the *PCK1*, *FBP1*, and *ICL1* genes decreased 5.1-fold, 2.0-fold, and 2.4-fold, respectively, compared with those in the wild-type strain, while their expression in YPD medium was unaffected by the deletion of the *PBP1* gene (Fig 2A–2C). Unlike microarray data, the mRNA level of *MLS1*, which codes another important enzyme in glyoxylate cycle, was not significantly decreased in the *pbp1Δ* mutant according to qRT-PCR result (Table 3, S1A Fig). In group 2, the mRNA levels of *COX10*, *COX11*, and *CYT2*, whose functions are related to cytochrome c, were significantly decreased compared to those in wild-type cells in YPGL medium (Fig 2D and 2F). The gene coding for cytochrome c oxidase subunit 8 (*COX8*) did not show a decreased expression in *pbp1Δ* mutant in YPGL (Table 3 and S1B Fig). Likewise, in the remaining groups, Pbp1 affected several specific genes functioning in mitochondrial ribosome large subunits (*MRPL4*, *MRPL3*, *MRPL17*, and *MRPL39*), mitochondrial ribosome small subunits (*MRPS35*, *MRP13*, *RSM25*, and *RSM27*), mitochondrial t-RNA synthetase (*MSY1*, *DIA4*, *SLM5*, *MSF1*, *ISM1*, and *MSW1*), mitochondria genome (*AIM33*, *AIM11*, and *AIM36*), and mitochondria matrix (*IBA57*, *MAM33*, and *FMC1*) (Table 3 and S1C–S1G Fig). In consistent with microarray data, the *AIM7* and *MCX1* mRNA levels were not significantly decreased in the *pbp1Δ* mutant according to qRT-PCR result (Table 3, S1F and S1G Fig). Interestingly, the mRNA level of *PBP1* is similar between YPD and YPGL medium (Table 2 and Fig 2G). This suggests that the *PBP1*-dependent change in the expression of those mentioned genes in YPGL medium does not result from a change in the transcript abundance of *PBP1*.

**Table 2. Comparative expression levels of selected genes from microarray data which are mentioned in the main text, *ACT1*, *PGK1*, and *PBP1* from microarray data.** The gene names that showed a decreased expression more than two-fold in *pbp1Δ* in YPGL are highlighted in underlined. The data show the relative microarray values of each gene obtained from mutant strains. These values were normalized against their corresponding wild-type value which reflects the fold change in expression (values in parenthesis).

| GENE | wild-type YPD | *pbp1Δ* YPD | wild-type YPGL | *pbp1Δ* YPGL |
|---|---|---|---|---|
| PCK1 | 78 (1) | 70 (0.90) | 5,951 (1) | 2,073 (0.35) |
| FBP1 | 52 (1) | 31 (0.60) | 5,158 (1) | 2,200 (0.43) |
| ICL1 | 71 (1) | 76 (1.07) | 1,920 (1) | 835 (0.43) |
| MLS1 | 24.3 (1) | 18.3 (0.75) | 2097.5 (1) | 655.2 (0.31) |
| COX10 | 361 (1) | 293 (0.81) | 840 (1) | 287 (0.34) |
| COX11 | 431 (1) | 383 (0.89) | 911 (1) | 511 (0.56) |
| CYT2 | 573.6 (1) | 426.9 (0.74) | 1277.9 (1) | 690.7 (0.54) |
| COX8 | 3527 (1) | 3245 (0.92) | 5669.2 (1) | 5570.2 (0.90) |
| MRPL4 | 275 (1) | 146 (0.53) | 766 (1) | 323 (0.42) |
| MRPL3 | 1258 (1) | 906 (0.72) | 1872.1 (1) | 964.6 (0.51) |
| MRPL17 | 909 (1) | 617 (0.68) | 1584.2 (1) | 825.9 (0.52) |
| MRPL39 | 1430.4 (1) | 1338.7 (0.94) | 2823 (1) | 2541.4 (0.90) |
| MRPS35 | 545.8 (1) | 389.1 (0.71) | 1283.7 (1) | 603.9 (0.47) |
| MRP13 | 786.5 (1) | 579.3 (0.73) | 1399.5 (1) | 707.9 (0.51) |
| RSM25 | 319.1 (1) | 232.6 (0.73) | 951.3 (1) | 484.8 (0.51) |
| RSM27 | 525.3 (1) | 471 (0.90) | 972.6 (1) | 894.3 (0.92) |
| GCV2 | 273.1 (1) | 329.4 (1.2) | 941.2 (1) | 451.2 (0.48) |
| GCV1 | 245.6 (1) | 239.3 (0.97) | 1005.4 (1) | 542.5 (0.54) |
| MSY1 | 375.7 (1) | 287.3 (0.76) | 1098.8 (1) | 481.4 (0.44) |
| DIA4 | 118.6 (1) | 85.8 (0.72) | 393.6 (1) | 182.1 (0.46) |
| SLM5 | 224.4 (1) | 165.2 (0.73) | 501.9 (1) | 241.1 (0.48) |
| MSF1 | 216.3 (1) | 167.9 (0.78) | 460.4 (1) | 237.7 (0.52) |
| ISM1 | 198.4 (1) | 165.9 (0.84) | 569.1 (1) | 315.8 (0.55) |
| MSW1 | 288.3 (1) | 268 (0.93) | 456.5 (1) | 378.4 (0.83) |

(*Continued*)

**Table 2.** (Continued)

| GENE | wild-type YPD | *pbp1Δ* YPD | wild-type YPGL | *pbp1Δ* YPGL |
|---|---|---|---|---|
| *AIM33* | 94 (1) | 91 (0.97) | 1658 (1) | 773 (0.47) |
| *AIM11* | 235.7 (1) | 164,4 (0,70) | 693.5 (1) | 410.1 (0.59) |
| *AIM36* | 649.9 (1) | 538.6 (0.83) | 1326.2 (1) | 791.4 (0.60) |
| *AIM7* | 1813.7 (1) | 1826.9 (1.0) | 2046.3 (1) | 2042.9 (1.0) |
| *IBA57* | 254.5 (1) | 191.9 (0.75) | 1091.1 (1) | 514.9 (0.47) |
| *MAM33* | 895.7 (1) | 564.7 (0.63) | 2610.3 (1) | 1481.8 (0.57) |
| *FMC1* | 524.6 (1) | 435.9 (0.83) | 1018.1 (1) | 679 (0.67) |
| *MCX1* | 703.2 (1) | 614.9 (0.87) | 580.8 (1) | 563.9 (0.97) |
| *ACT1* (control) | 10,711 (1) | 9.845 (0.92) | 8,279 (1) | 8,040 (0.97) |
| *PGK1* (control) | 10,819 (1) | 10,212 (0.94) | 8,298 (1) | 8,268 (1) |
| *PBP1* | 693 (1) | 7.7 (0.01) | 953 (1) | 3 (0) |

To confirm that the decreased expression of the above genes is caused by the *pbp1Δ* mutation, we examined whether the *PBP1* plasmid rescues the decreased expression (Fig 3). The decreased expression of *COX10*, *COX11*, and *CYT2* in the *pbp1Δ* mutant was efficiently rescued by the *PBP1* plasmid. However, in this growth condition using -UraGL, expression of *PCK1*, *FBP1*, and *ICL1* was not decreased in the *pbp1Δ* mutant compared to wild-type cells. We repeated to examine the gene expression in the *pbp1Δ* mutant cultivated in different media, such as YPGL and -UraGL, and found that the difference in the expression of *PCK1*, *FBP*, and *ICL1* between wild-type and *pbp1Δ* was observed in YPGL, not in synthetic medium. As shown in Fig 1C, the slow growth of the *pbp1Δ* mutant on -UraGL plate was efficiently complemented by the *PBP1* plasmid. Therefore, the slow growth of the *pbp1Δ* mutant on -UraGL seems to mainly depend on the decreased expression of the genes involved in mitochondrial function, but not in gluconeogenesis pathway.

**Table 3. Summary of RT-PCR result, genes are classified into 7 groups based on gene ontology.**

| Group No. | Function | Genes which showed a DECREASED expression in *pbp1Δ* mutant in RT-PCR | Genes which showed an UNCHANGED expression *pbp1Δ* mutant in RT-PCR |
|---|---|---|---|
| 1 | Gluconeogenesis | *PCK1, FBP1, ICL1* | *MLS1* |
| 2 | Cytochrome c | *COX10, COX11, CYT2* | *COX8* |
| 3 | Mitochondria1 ribosomal large subunit | *MRPL4, MRPL3, MRPL17, MRPL39* | |
| 4 | Mitochondria1 ribosomal small subunit | *MRPS35, MRP13. RSM25, RSM27* | |
| 5 | Mitochondrial t-RNA synthetase | *MSY1, DIA4, SLM5, MSF1, ISM1, MSW1* | |
| 6 | Mitochondria genome | *AIM33, AIM11, AIM36* | *AIM7* |
| 7 | Mitochondria matrix | *IBA57, MAM33, FMC1* | *MCX1* |

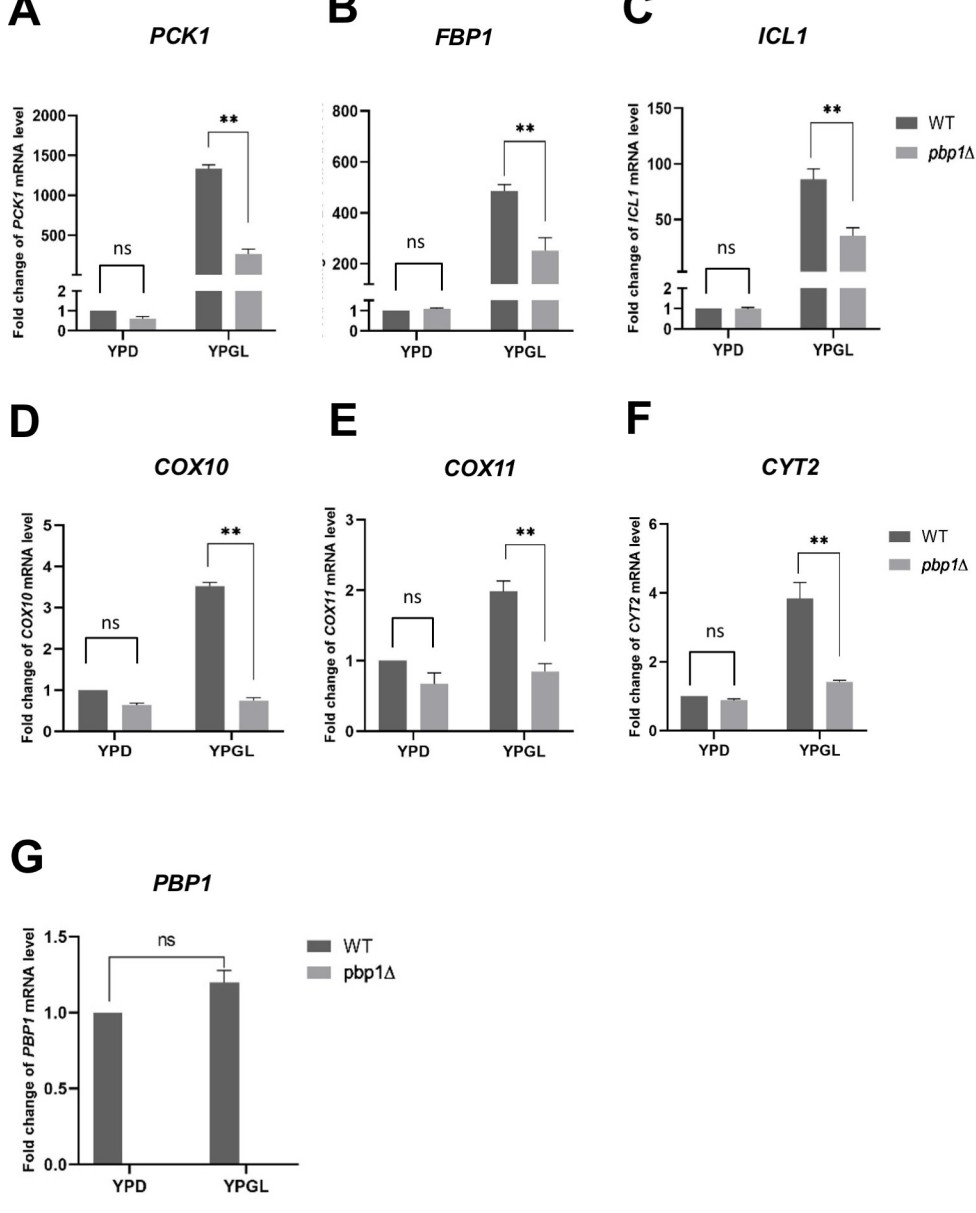

**Fig 2. Expression of *PCK1, FBP1, ICL1, COX10, COX11, CYT2,* and *PBP1* in wild-type and *pbp1Δ* mutant cells cultured in YPD and YPGL media.** The mRNA levels of (A) *PCK1*, (B) *FBP1*, (C) *ICL1*, (D) *COX10*, (E) *COX11*, (F) *CYT2*, and (G) *PBP1* in wild-type strain (WT) and *pbp1Δ* mutant strain growing in YPD and YPGL media. mRNA levels were quantified by qRT-PCR analysis, and the relative mRNA levels were calculated using $2^{-\Delta\Delta Ct}$ method normalized to *ACT1* reference gene. The data show mean ± SEM (n = 3) of fold change of mRNA level from wild-type cells at 4 h of culture in YPD. ns (not significant), *P < 0.05, **P < 0.01 as determined by Tukey's test.

## Pbp1 affects the expression of genes involved in gluconeogenesis through their promoters but controls the expression of genes related to cytochrome c not through their promoters

Our microarray data revealed that Pbp1 influences a range of mRNA involved in the gluconeo-genesis pathway and mitochondrial function. The drop in the mRNA level might be caused by either a reduction in transcription or a decrease in mRNA stability. To investigate how Pbp1

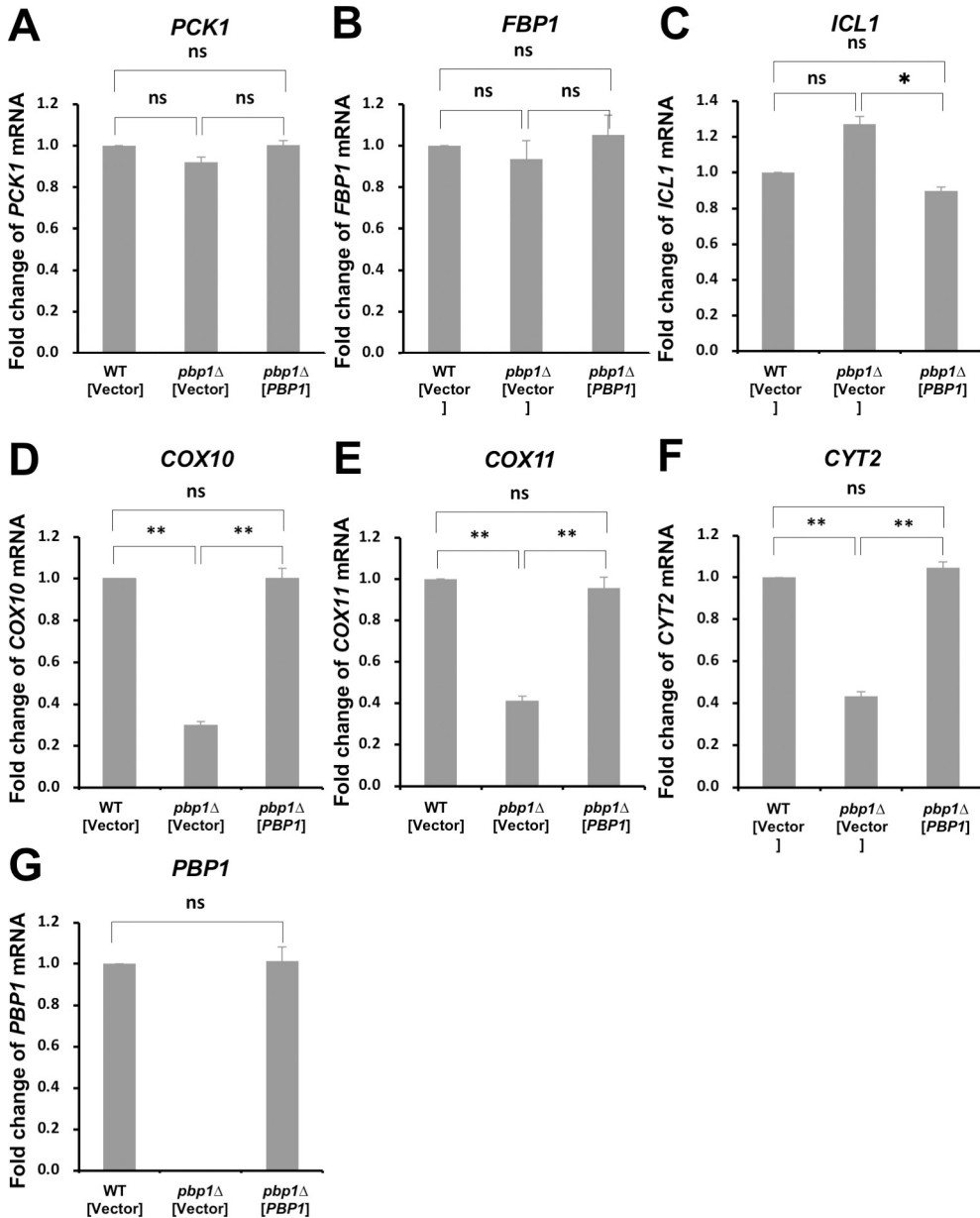

**Fig 3. Expression of *PCK1*, *FBP1*, *COX10*, and *COX11* in the *pbp1Δ* mutant cells harboring vector or YCpPBP1 plasmid.** The mRNA levels of (A) *PCK1*, (B) *FBP1*, (C) *ICL1*, (D) *COX10*, (E) *COX11*, (F) *CYT2*, and (G) *PBP1* in wild-type strain harboring YCplac33 (WT [vector]), the *pbp1Δ* mutant harboring YCplac33 (*pbp1Δ* [vector]), and the *pbp1Δ* mutant harboring YCpPBP1 (*pbp1Δ* [YCpPBP1]) growing in -UraGL medium. mRNA levels were quantified by qRT-PCR analysis, and the relative mRNA levels were calculated using $2^{-\Delta\Delta Ct}$ method normalized to *SCR1* reference gene. The data show mean ± SEM (n = 3) of fold change of mRNA level from wild-type cells at 4 h of culture in -UraGL. ns (not significant), *P < 0.05, **P < 0.01 as determined by Tukey's test.

controls the mRNA levels in YPGL medium, we constructed reporter plasmids that harbor the *GFP* gene driven by the promoter of each gene and transformed them into wild-type and *pbp1Δ* mutant cells on -UraGlu plate (Fig 4). We cultured these Ura+ transformants in the YPGL medium containing glycerol and lactate, and examined the mRNA levels of *GFP* driven by the promoter of *PCK1*, *FBP1*, and *ICL1*, together with the mRNA levels of the

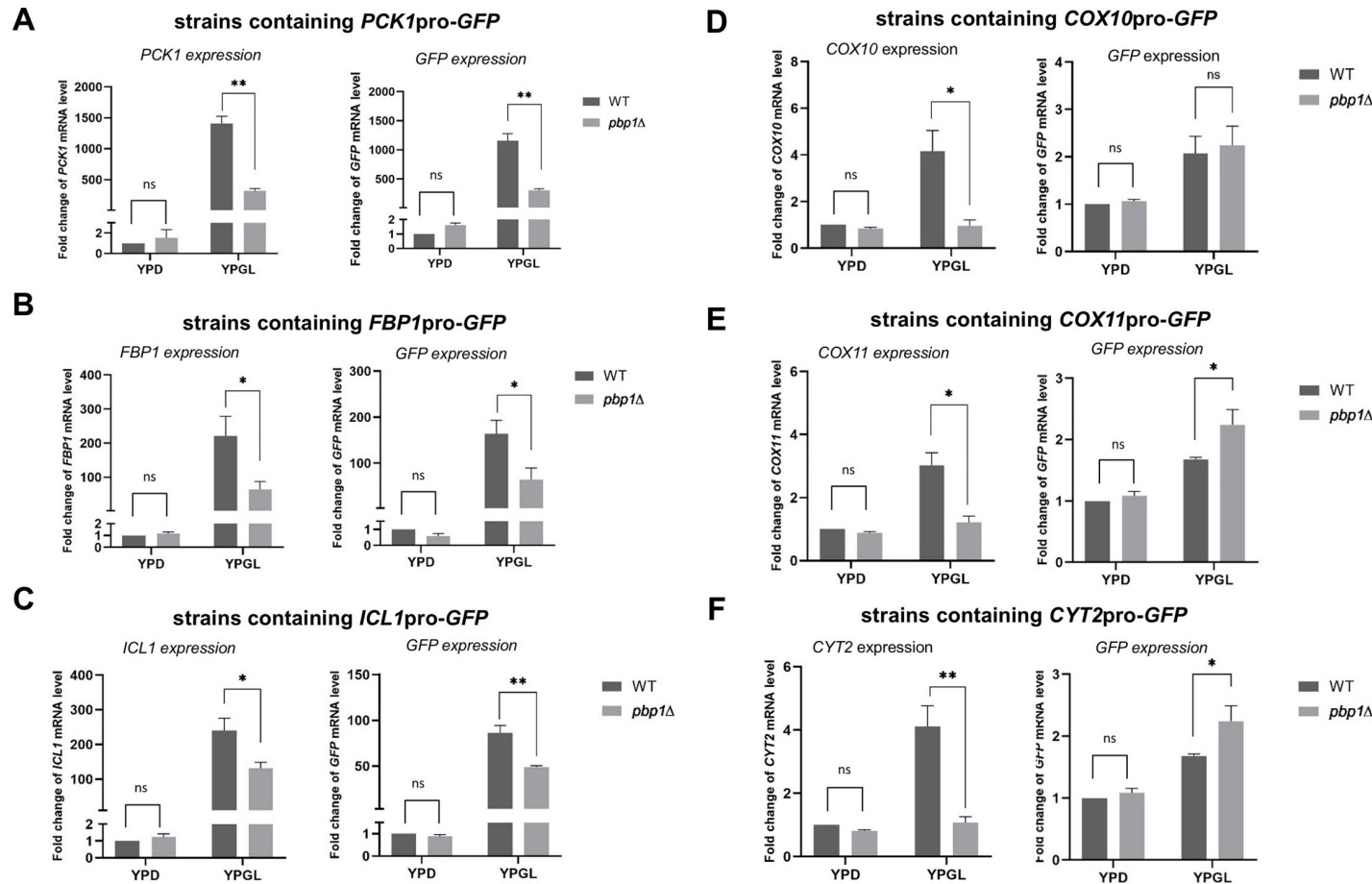

**Fig 4. Expression of *GFP* reporter genes driven by the promoters of Pbp1 targets.** All the right panels display the mRNA levels of *GFP* gene driven by the promoter of (A) *PCK1*, (B) *FBP1*, (C) *ICL1*, (D) *COX10*, (E) *COX11*, and (F) *CYT2* in wild-type strain (WT) and *pbp1Δ* mutant strain harboring the GFP reporter plasmid growing in YPD and YPGL media. The left panels display the mRNA level of the corresponding endogenous genes in the same strain as the right panels. The total mRNA is used to measure mRNA level of both endogenous genes and *GFP* gene. mRNA levels were quantified by qRT-PCR analysis, and the relative mRNA levels were calculated using $2^{-\Delta\Delta Ct}$ method normalized to *ACT1* reference gene. The data show mean ± SEM (n = 3) of fold change of mRNA level from wild-type cells at 4 h of culture in YPD. ns (not significant), *P < 0.05, **P < 0.01 as determined by Tukey's test.

corresponding endogenous genes (Fig 4A–4C). Since the GFP reporter plasmid is YCp-based plasmid, most cells harbored the plasmid in YPGL culture. In *pbp1Δ* mutant cells, the *GFP* expression from *PCK1*, *FBP1*, and *ICL1* promoters was significantly decreased compared with wild-type strain. This suggests that Pbp1 regulates the expression of *PCK1*, *FBP1*, and *ICL1* through their promoter. On the other hand, the *pbp1Δ* mutation did not significantly reduce *GFP* expression from the promoter of *COX10*, *COX11*, and *CYT2* in YPGL medium, and even slightly increase *GFP* mRNA level in the case of *COX11* and *CYT2* (Fig 4D–4F). Thus, Pbp1 regulates the expression of genes categorized in group 2 (cytochrome c related genes), not through their promoter. Similarly, the *GFP* reporter assay indicated that the promoter is not involved in the Pbp1-mediated regulation of the genes representing the remaining five groups (S2A–S2E Fig).

We, therefore, concluded that the mechanism by which Pbp1 controls the mRNA level of the genes involved in the gluconeogenesis pathway and the genes related to mitochondrial functions are different. While Pbp1 regulates the expression of gluconeogenesis genes through their promoters, Pbp1 controls the expression of mitochondria genes not through the promoters of these genes.

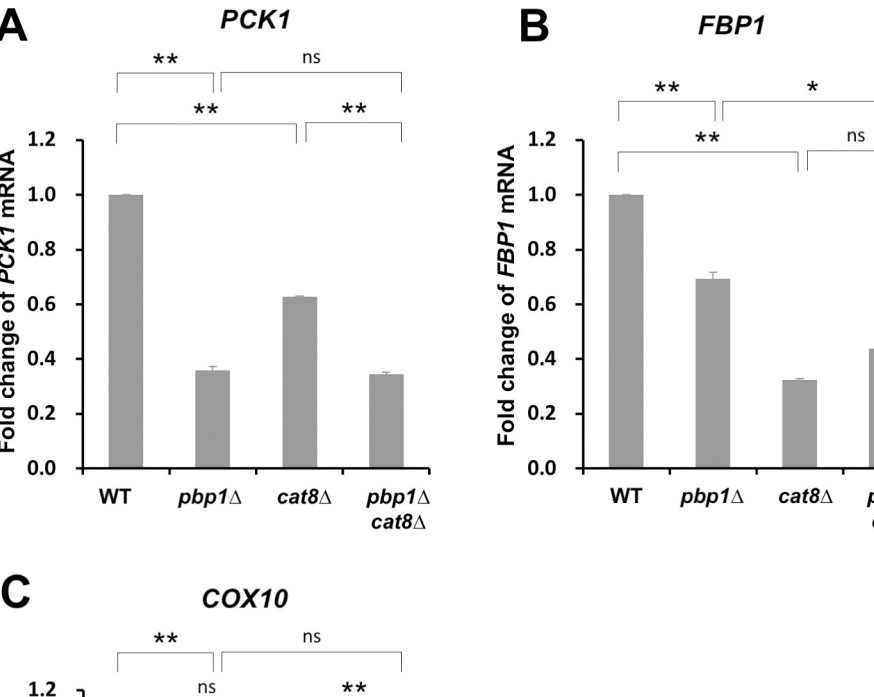

**Fig 5. Expression of *PCK1*, *FBP1*, and *COX10* mRNA in wild-type, *pbp1Δ*, *cat8Δ*, and *pbp1Δ cat8Δ* mutants cultured in YPGL medium.** The mRNA levels of (A) *PCK1*, (B) *FBP1*, and (C) *COX10* in wild-type (WT), *pbp1Δ*, *cat8Δ*, and *pbp1Δ cat8Δ* mutants growing and YPGL medium. mRNA levels were quantified by qRT-PCR analysis, and the relative mRNA levels were calculated using $2^{-\Delta\Delta Ct}$ method normalized to *SCR1* reference gene. The data show mean ± SEM (n = 3) of fold change of mRNA level from wild-type cells at 4 h of culture in YPGL. ns (not significant), *P < 0.05, **P < 0.01 as determined by Tukey's test.

## Pbp1 and TORC1 act in the same pathway to control the expression of *PCK1*

The *GFP* reporter assay revealed that Pbp1 controls the expression of *PCK1*, *FBP1*, and *ICL1* through their promoter. Cat8 is a transcriptional regulator known to function in catabolite derepression, releasing the inhibition of various genes essential for non-fermentable carbon utilization [35]. Cells lacking Cat8 are not capable of growing on non-fermentable carbon sources [35]. It has also been reported that Cat8 targets gluconeogenesis genes such as *PCK1* and *FBP1* [36]. To investigate whether Pbp1 regulates *PCK1* and *FBP1* through Cat8, a *pbp1Δ cat8Δ* double mutant was prepared and analyzed. The mRNA levels of *PCK1* and *FBP1* were lower in the *cat8Δ* single mutant compared to the wild-type cells (Fig 5A and 5B). We did not

observe such a reduction in the *COX10* mRNA level in *cat8Δ* mutant (Fig 5C). These results confirmed that Cat8 plays a role in activating *PCK1* and *FBP1*, but not *COX11*. In the case of *PCK1*, the *pbp1Δ* mutation resulted in a significant decrease in mRNA level and the addition of *cat8Δ* mutation resulted in no significant change in mRNA level. In *FBP1*, the *cat8Δ* mutant exhibited a noticeable decrease in mRNA level. However, no significant difference was observed between the *cat8Δ* single mutant and the *pbp1Δ cat8Δ* double mutant, thus intriguing to investigate whether Cat8 involves in the regulatory activity of Pbp1 on cell growth in YPGL medium.

Since Cat8 acts on transcriptional activation of *PCK1* and *FBP1*, it was considered that the growth deficiency observed in the *pbp1Δ* mutant strain in YPGL medium conditions was further exacerbated by the *cat8Δ* mutation, and the growth was confirmed. The growth of the *pbp1Δ cat8Δ* double mutant was worse than that of the *pbp1Δ* single mutant (Fig 6A). In addition, growth in YPD medium did not change in both the *cat8Δ* mutant and the *pbp1Δ cat8Δ* double mutant (Fig 6A). We also examined the growth in liquid media and found that the *pbp1Δ cat8Δ* double mutant showed slower growth than the *pbp1Δ* single mutant in YPGL medium (Fig 6B). From the above results, it was suggested that the regulation of Pbp1 is not mediated by the transcriptional activator Cat8 but is regulated in parallel.

Previous studies have identified that Pbp1 and its ortholog Ataxin-2 interact and recruit TORC1 into stress granules to downregulate TOR signaling during nutrient depletion [23, 24]. To examine whether Pbp1 regulates the transcription of *PCK1*, together with TORC1, we deleted the *TOR1* gene coding for one of the catalytic subunits of TORC1 and constructed the *pbp1Δ tor1Δ* double mutant. As shown in Fig 6, the *tor1Δ* mutation decreased the level of *PCK1* mRNA in YPGL media. This suggests that Tor1 is a positive regulator for *PCK1* expression. Also, deleting both the *PBP1* and *TOR1* had no synthetic decrease in *PCK1* mRNA level (Fig 7), raising the possibility that Pbp1 and Tor1 act in the same pathway to control the transcription of *PCK1*.

## The *COX10* 3′ UTR is necessary but insufficient for Pbp1-mediated regulation of *COX10* expression in YPGL

The above data suggested that, under non-fermentative growth, Pbp1 regulates mitochondrial-related genes (*COX10*, *COX11*, *AIM33*, etc) without modulating their promoter activity. We have previously reported that Pbp1, together with Mkt1, regulates *HO* expression at the posttranscriptional level via the *HO* 3′ UTR [15]. Thus, we investigated whether the 3′ UTR is important for Pbp1 to control the mRNA levels of *COX10* and *COX11*. We constructed the *COX10-ADH1* 3'UTR and the *COX11-ADH1* 3'UTR by replacing the endogenous *COX10* 3′ UTR or *COX11* 3′ UTR (the sequence containing 500 bp downstream of the stop codon) with *ADH1* 3′ UTR and compared their mRNA levels. This modification did not result in a clear effect on *COX10* and *COX11* mRNA levels in *pbp1Δ* comparing to wild-type in YPD medium (Fig 8A and 8B). In contrary, the reduction of *COX10* and *COX11* mRNA levels in the *pbp1Δ* mutant in YPGL was significantly ameliorated in the *COX10-ADH1* 3'UTR and the *COX11-ADH1* 3'UTR constructs (Fig 8A and 8B). These data suggest that the 3′ UTRs of *COX10* and *COX11* are necessary for the regulation of their mRNA levels by Pbp1 in YPGL.

To further examine the role of 3′ UTR in the regulation of *COX10* and *COX11*, we constructed the plasmids which contain *COX10* promoter-GFP-*COX10* 3'UTR and *COX11* promoter-GFP-*COX11* 3'UTR. Although the *GFP* mRNA level from the *COX10* promoter-GFP-*COX10* 3'UTR plasmid was slightly decreased in the *pbp1Δ* mutant compared to that in wild-type cells when cultivated in YPGL medium, the reduction in mRNA level observed in the *GFP* reporter gene was not as strong as the one observed in endogenous *COX10* mRNA of

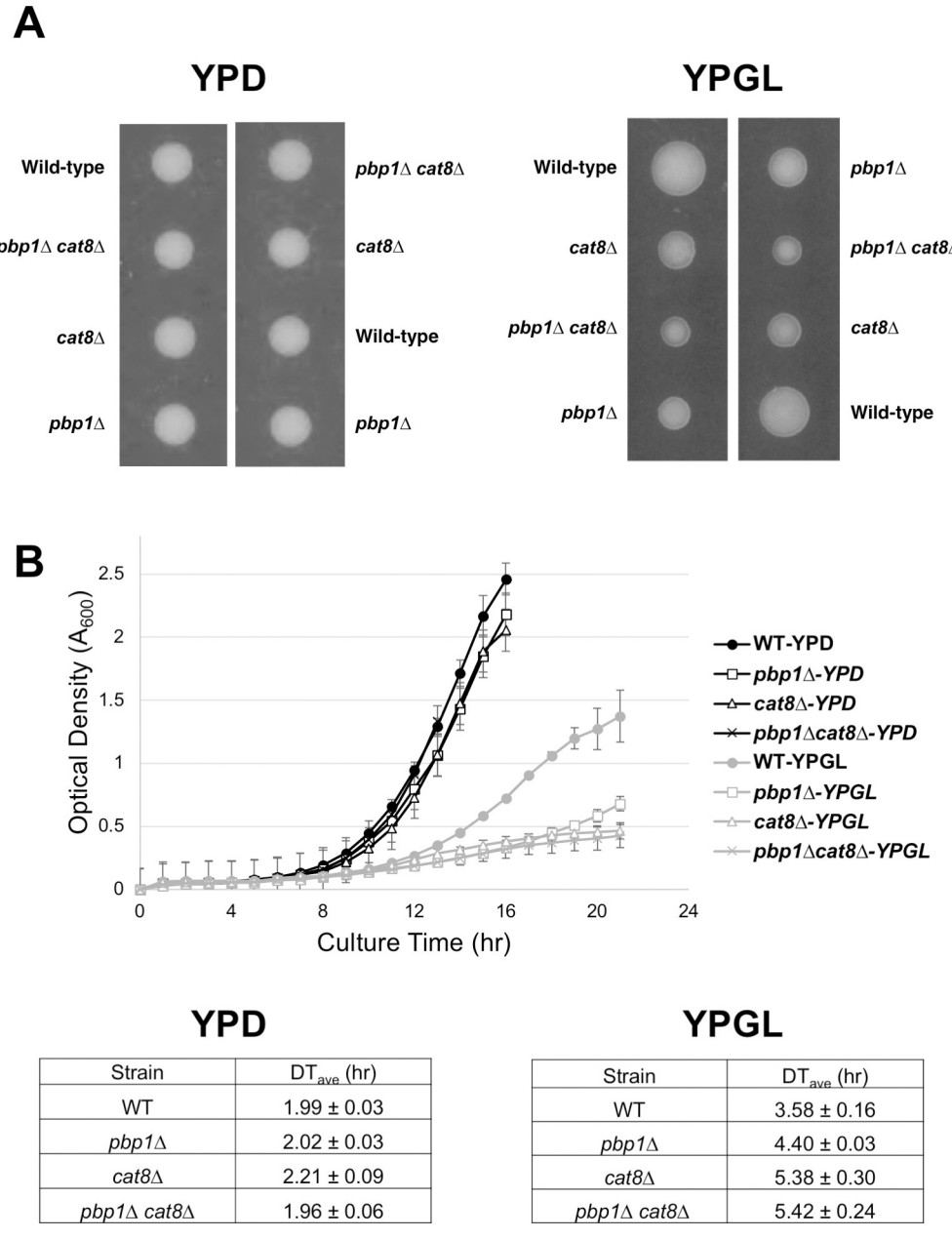

**Fig 6. The *cat8Δ* mutation exacerbated the growth of the *pbp1Δ* mutant strain.** (A) The cells that are heterozygous for the *cat8Δ* and *pbp1Δ* alleles, were sporulated, and tetrads were dissected onto YPD and YPGL plates. Growth after 3 days (YPD) and 8 days (YPGL) at 30°C is shown. More than 20 tetrads were dissected, and representative data were shown. (B) The growth curves of WT, *pbp1Δ*, *cat8Δ*, and *cat8Δ pbp1Δ* mutants in YPD and YPGL media. The strains were pre-cultured overnight in YPD and then transferred into fresh YPD and YPGL to grow at 30°C. We measured the rate of transmitted light (950 nm) volume at 60-min intervals using an ODBox-C/ODMonitor system (Taitec Corp., Saitama, Japan). The transmission rate was converted into optical density (OD), and the average OD over 60 min was calculated for each sample. The data of doubling time show mean ± SEM (n = 3).

the same strain (Fig 9A). The *GFP* mRNA level from the *COX11* promoter-GFP-*COX11* 3'UTR plasmid was not decreased in the *pbp1Δ* mutant (S3 Fig). Thus, the 3′ UTRs of *COX10* and *COX11* are not sufficient for the regulation of their mRNA levels by Pbp1 in YPGL.

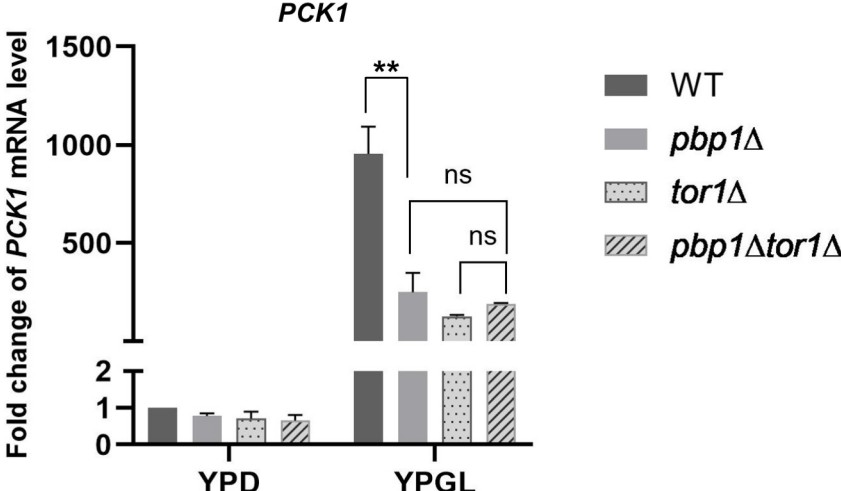

**Fig 7. Expression of *PCK1* mRNA in wild-type strain (WT), *pbp1Δ*, *tor1Δ*, and *pbp1Δ tor1Δ* mutants growing in YPD and YPGL media.** mRNA levels were quantified by qRT-PCR analysis, and the relative mRNA levels were calculated using $2^{-\Delta\Delta Ct}$ method normalized to *ACT1* reference gene. The data show mean ± SEM (n = 3) of fold change of mRNA level from wild-type cells at 4 h of culture in YPD. ns (not significant), *P < 0.05, **P < 0.01 as determined by Tukey's test.

Lastly, we constructed the *COX10* promoter-*COX10* coding sequence-*COX10* 3'UTR plasmid and introduced it into the *cox10Δ* and *pbp1Δ cox10Δ* cells. The expression of *COX10* mRNA from the plasmid was significantly decreased in the *pbp1Δ cox10Δ* mutant comparing to that in the *cox10Δ* mutant (Fig 9B), suggesting that the *COX10* coding sequence is also necessary for Pbp1-mediated regulation.

## *COX10*, *COX11*, and *AIM33* mRNA are not regulated by Pbp1 via the Pan2-Pan3 complex and the Ccr4-Not complex

We next investigated the details of the regulation of *COX10*, *COX11*, and *AIM33*. The poly (A) tail degrading enzyme Pan2-Pan3 complex has been reported to be inhibited by Pbp1 [19]. To examine whether the Pan2-Pan3 complex controls *COX10*, *COX11*, and *AIM33* mRNA, we prepared the *pbp1Δ pan2Δ* double mutant and then performed qRT-PCR analysis. The *COX10* mRNA level was decreased in the *pbp1Δ* single mutant compared to the wild-type cells, and this reduction was not restored by the *pan2Δ* mutation (Fig 10A). Similar results were obtained at the *COX11* and *AIM33* mRNA levels (Fig 10B and 10C). These results suggest that Pbp1 does not regulate the expression of *COX10*, *COX11*, and *AIM33* mRNAs through the Pan2-Pan3 complex.

The Ccr4-Not complex is another major enzyme for poly(A) degradation [37, 38]. To investigate whether Pbp1 performs its functions with the Ccr4-Not complex in the YPGL medium, we created and analyzed the *pbp1Δ ccr4Δ* double mutant. The *COX10* mRNA level was decreased in the *pbp1Δ* single mutant compared to the wild-type cells, and this reduction was not restored by the *ccr4Δ* mutation (Fig 11A). Similar results were obtained at the *COX11* and *AIM33* mRNA levels (Fig 11B and 11C). These results suggest that Pbp1 does not regulate the expression of *COX10*, *COX11*, and *AIM33* mRNAs through the Ccr4-Not complex.

## Pbp1 stabilizes the mRNA of *COX10*, *COX11*, and *AIM33* genes by inhibiting decapping

We next focused on the factors involved in the control of decapping and 5'-3' exonuclease (Xrn1, Dhh1, Dcp1-Dcp2 complex), which play an important role in regulating mRNA

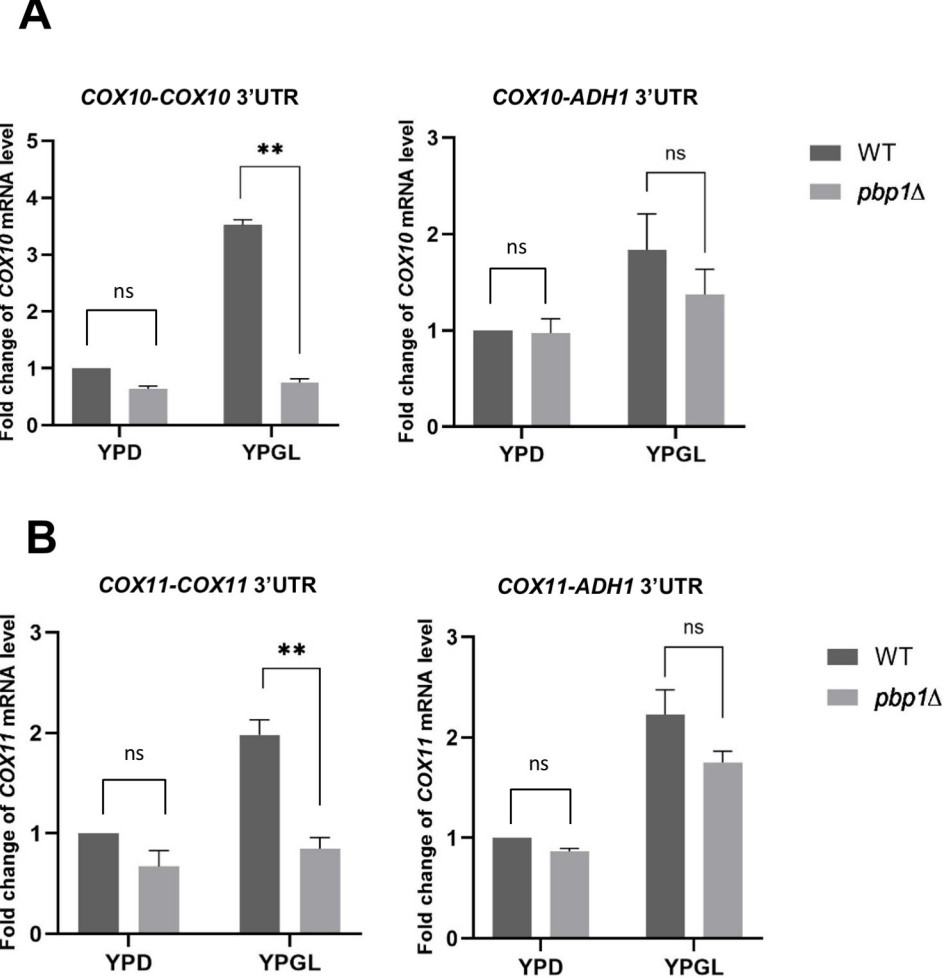

**Fig 8. The importance of *COX10* 3'UTR and *COX11* 3'UTR to the regulatory effect of Pbp1 on *COX10* and *COX11* mRNA level in YPGL media.** The mRNA levels of (A) *COX10* and (B) *COX11* gene with endogenous 3'UTR (left panel) and with *ADH1* terminator (*ADH1* ter) (right panel) in wild-type strain (WT) and *pbp1Δ* mutant strain growing in YPD and YPGL media. mRNA levels were quantified by qRT-PCR analysis, and the relative mRNA levels were calculated using $2^{-\Delta\Delta Ct}$ method normalized to *ACT1* reference gene. The data show mean ± SEM (n = 3) of fold change of mRNA level from wild-type cells at 4 h of culture in YPD. ns (not significant), *P < 0.05, **P < 0.01 as determined by Tukey's test.

stability [38]. To investigate whether Pbp1 in the YPGL medium acts with Xrn1 5'-3' exonuclease, we prepared and analyzed the *pbp1Δ xrn1Δ* double mutant strain. The *COX10* mRNA level was decreased in the *pbp1Δ* single mutant compared to the wild-type cells, and this reduction was suppressed by the *xrn1Δ* mutation (Fig 12A). Similar results were observed at the *COX11* and *AIM33* mRNA levels (Fig 12B and 12C).

Dhh1 is a decapping activator [38]. To investigate whether Pbp1 in YPGL medium acts on Dhh1, a double mutant of *pbp1Δ dhh1Δ* was prepared and the same analysis was performed. As a result, the *COX10* mRNA level in the *pbp1Δ* single mutant, which decreased compared to the wild-type, was restored by the *dhh1Δ* mutation (Fig 13A). A similar result was observed at *COX11* mRNA levels (Fig 13B). The decreased level of *AIM33* mRNA was only weakly recovered by the *dhh1Δ* mutation (Fig 13C).

The Dcp1-Dcp2 complex is a decapping factor [38]. Since the *dcp1* deficiency is lethal in the strain used in our laboratory, it was not possible to prepare a *pbp1Δ dcp1Δ* double mutant

# A

### strains containing *COX10*pro-*GFP-COX10* 3'UTR

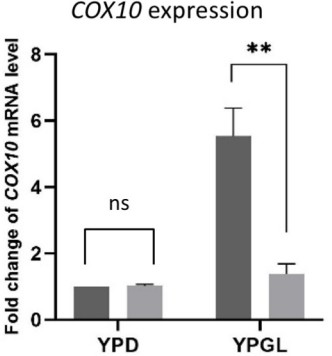

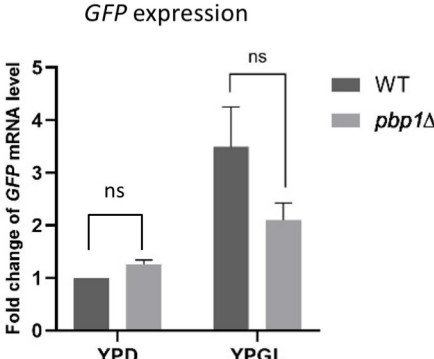

# B

### strains containing *COX10*pro-*COX10-COX10* 3'UTR

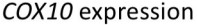

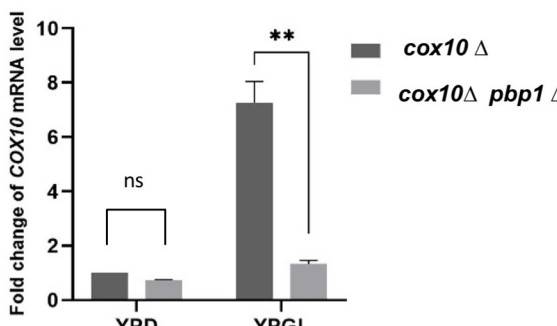

**Fig 9. The importance of *COX10* 3'UTR and coding sequence to Pbp1 effect on *COX10* mRNA level in YPGL media.** (A) The level of *GFP* mRNA driven by *COX10* promoter with *ADH1* terminator (left panel) and with *COX10* 3'UTR (right panel), (B) The level of *COX10* mRNA in *cox10Δ* and *pbp1Δ cox10Δ* harboring *COX10* whole gene in YCplac33 vector. mRNA levels were quantified by qRT-PCR analysis, and the relative mRNA levels were calculated using $2^{-\Delta\Delta Ct}$ method normalized to *ACT1* reference gene. The data show mean ± SEM (n = 3) of fold change of mRNA level from wild-type cells at 4 h of culture in YPGL. ns (not significant), $^{*}$P < 0.05, $^{**}$P < 0.01 as determined by Tukey's test.

strain. Therefore, we implemented an Auxin-Inducible degron (AID) system which allows a rapid depletion of Dcp1 [39]. We fused the AID tag to the Dcp1 protein and introduced the ubiquitin ligase subunit transport inhibitor response 1 (TIR1) into the yeast cells. The addition of 1-Naphthaleneacetic acid (NAA), which is a synthetic plant hormone in the auxin family, activated the ubiquitin ligase SCF-TIR1 complex which in turn degraded AID tagged Dcp1 protein. The efficacy of the AID system was confirmed by Western blotting. In strains where TIR1 was not introduced, Dcp1 was not degraded upon NAA addition, whereas in cells having TIR1, almost no Dcp1 band was visible one hour after NAA was added (S4 Fig). Using this AID system, we investigated whether Pbp1 acts on the Dcp1-Dcp2 complex during non-fermentative growth. The qRT-PCR revealed that the decrease in *COX10* mRNA level caused by

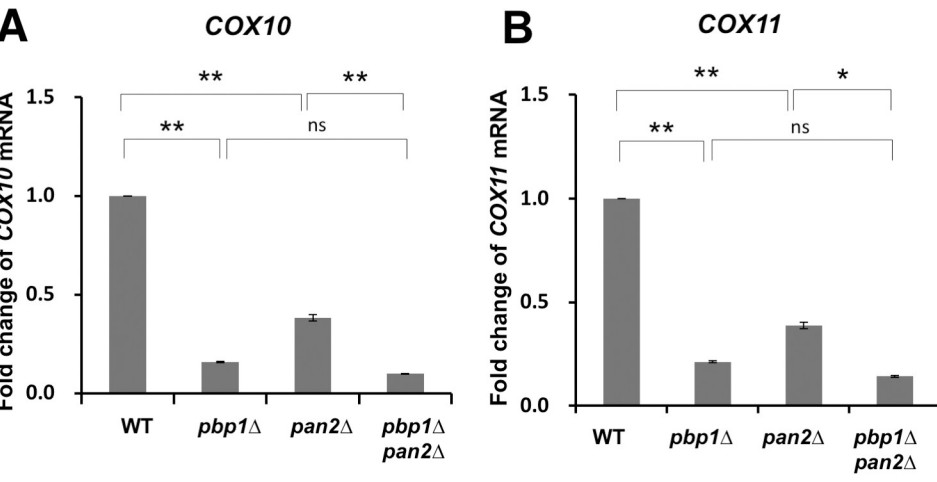

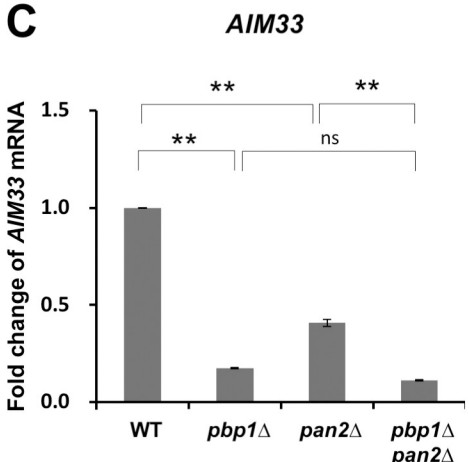

**Fig 10. Expression of *COX10*, *COX11*, and *AIM33* mRNA in wild-type strain, *pbp1Δ*, *pan2Δ*, and *pbp1Δ pan2Δ* mutants growing in YPGL medium.** The mRNA levels of (A) *COX10*, (B) *COX11*, and (C) *AIM33* in wild-type (WT), *pbp1Δ*, *pan2Δ*, and *pbp1Δ pan2Δ* mutants growing and YPGL medium. mRNA levels were quantified by qRT-PCR analysis, and the relative mRNA levels were calculated using $2^{-\Delta\Delta Ct}$ method normalized to *SCR1* reference gene. The data show mean ± SEM (n = 3) of fold change of mRNA level from wild-type cells at 4 h of culture in YPGL. ns (not significant), *P < 0.05, **P < 0.01 as determined by Tukey's test.

the deletion of *PBP1* gene was ameliorated after the degradation of Dcp1 (Fig 14A). Similar results were observed for the *COX11* and *AIM33* mRNA levels (Fig 14B and 14C).

From the above results, Pbp1 can stabilize *COX10*, *COX11*, and *AIM33* mRNAs by inhibiting decapping factors rather than acting on deadenylases, the Ccr4-Not and Pan2-Pan3 complexes.

## Discussion

### Pbp1 has additional functions in non-fermentable carbon source containing media

Gene expression can be regulated through the alteration in the transcriptional level, the rate of protein synthesis, or post-translational modifications [1, 5, 6]. Here we show that the strain lacking Pbp1 displayed slower growth in YPGL medium compared to the wild-type strain

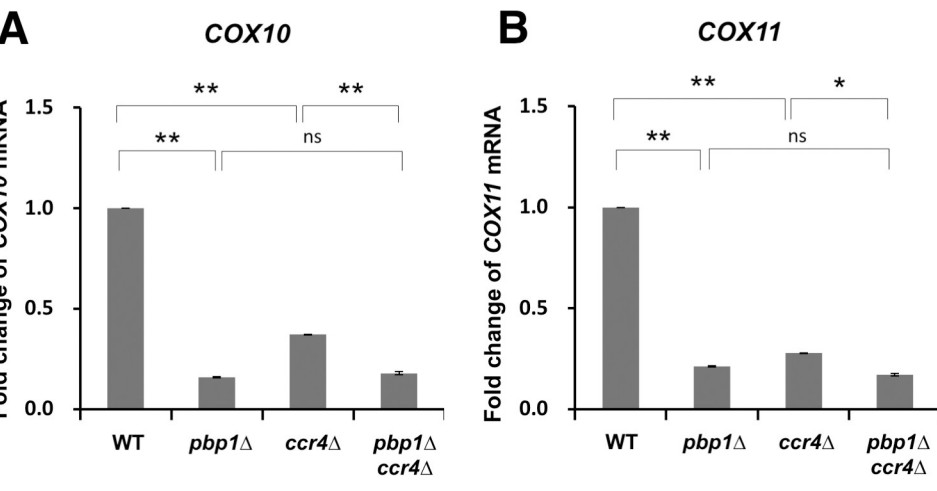

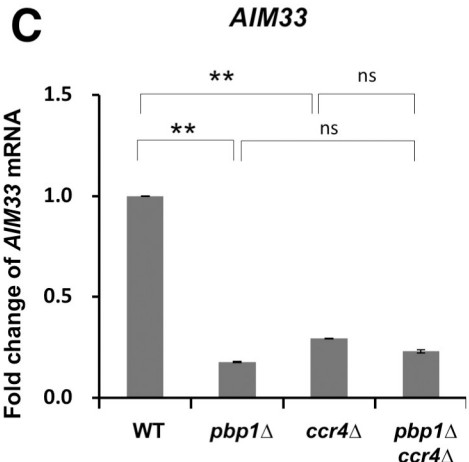

**Fig 11. Expression of *COX10*, *COX11*, and *AIM33* mRNA in wild-type strain, *pbp1Δ*, *ccr4Δ*, and *pbp1Δ ccr4Δ* mutants growing in YPGL medium.** The mRNA levels of (A) *COX10*, (B) *COX11*, and (C) *AIM33* in wild-type strain (WT), *pbp1Δ*, *ccr4Δ*, and *pbp1Δ ccr4Δ* mutants growing and YPGL medium. mRNA levels were quantified by qRT-PCR analysis, and the relative mRNA levels were calculated using $2^{-\Delta\Delta Ct}$ method normalized to *SCR1* reference gene. The data show mean ± SEM (n = 3) of fold change of mRNA level from wild-type cells at 4 h of culture in YPGL. ns (not significant), $^*P < 0.05$, $^{**}P < 0.01$ as determined by Tukey's test.

while it grew similar to the wild-type in YPD medium. This implies that Pbp1 is specifically required for cell growth on non-fermentable carbon source medium. Microarray data and RT-PCR results revealed that 43 and 24 transcripts are upregulated and downregulated, respectively, in the *pbp1Δ* mutant strain cultivated in YPGL medium compared to in wild-type strain. As the expression level of *PBP1* is similar between YPD and YPGL (Fig 2G), the change in Pbp1 function does not result from a change in the abundance of its transcripts. Rather, Pbp1 might have functions in non-fermentable carbon source-containing medium to control gene expression. How does Pbp1 gain those functions in non-fermentable carbon source-containing medium? Pbp1 is phosphorylated by Psk1 upon glucose deprivation and subsequently inhibits TORC1 through sequestering it to stress granules [23, 26]. The TORC1 complex is known as the master regulator that transcriptionally fine-tunes gene expression in responses to changes in nutrient availability [40–42]. Though Pbp1 is known as a post-transcriptional regulators, our results suggest that Pbp1 controls the gluconeogenesis genes through their

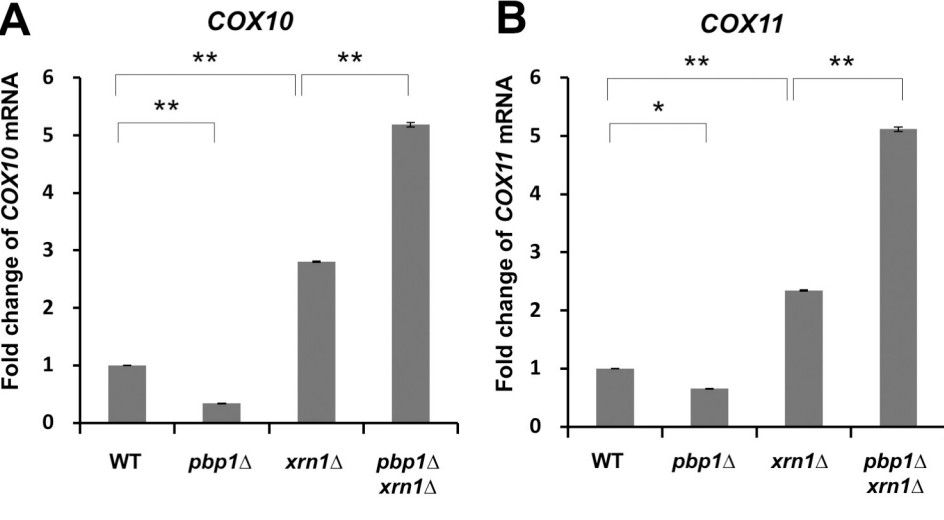

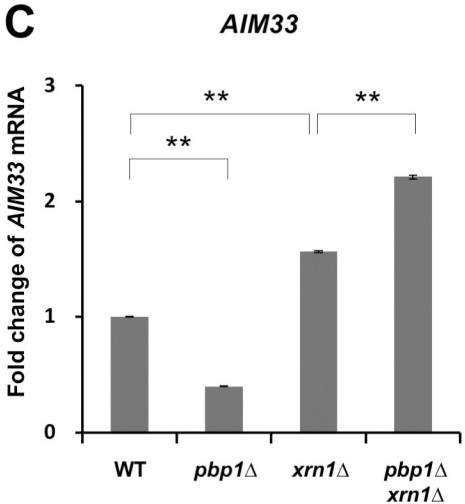

**Fig 12. Expression of *COX10*, *COX11*, and *AIM33* mRNA in wild-type strain, *pbp1Δ*, *xrn1Δ*, and *pbp1 Δxrn1Δ* mutants growing in YPGL medium.** The mRNA levels of (A) *COX10*, (B) *COX11*, and (C) *AIM33* in wild-type strain (WT), *pbp1Δ*, *xrn1Δ*, and *pbp1Δ xrn1Δ* mutants growing in YPGL medium. mRNA levels were quantified by qRT-PCR analysis, and the relative mRNA levels were calculated using $2^{-\Delta\Delta Ct}$ method normalized to *SCR1* reference gene. The data show mean ± SEM (n = 3) of fold change of mRNA level from wild-type cells at 4 h of culture in YPGL. ns (not significant), *P < 0.05, **P < 0.01 as determined by Tukey's test.

promoter. Thus we tried to decipher if TORC1 serves as the link for the regulation of Pbp1 on gluconeogenesis genes. Our data confirm that Tor1, the major subunit of TORC1 complex, positively contributes to the expression of *PCK1* in non-fermentable medium and suggest that Pbp1 might act in the same pathway as TORC1 and upstream TORC1 to regulate the transcription of *PCK1* in YPGL medium (Fig 7). To understand the mechanism involving Pbp1 in non-fermentable medium, we then examined the phosphorylation patterns of Pbp1. However, we were not able to find the significant changes in the phosphorylation status of Pbp1 between in YPD and YPGL media (unpublished data). Thus, the upstream signal of Pbp1 in YPGL medium remains elusive. In glucose starvation media, Pbp1 localizes to stress granules where it exhibits a modification in morphology [22, 25]. Whether these changes in localization and configuration of Pbp1 are attributable to its functions in YPGL is open for further studies.

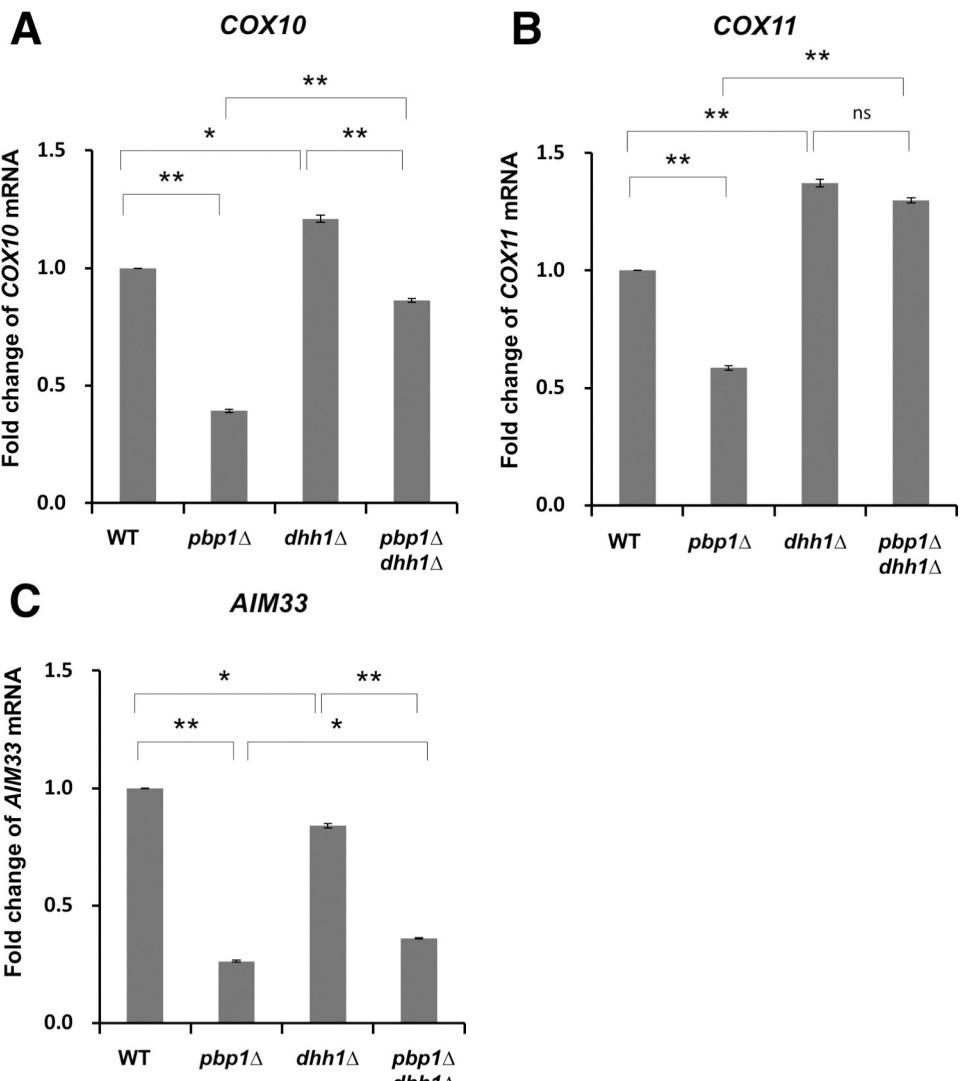

**Fig 13. Expression of *COX10*, *COX11*, and *AIM33* mRNA in wild-type strain, *pbp1Δ*, *dhh1Δ*, and *pbp1Δ dhh1Δ* mutants growing in YPGL medium.** The mRNA levels of (A) *COX10*, (B) *COX11*, and (C) *AIM33* in wild-type (WT), *pbp1Δ*, *dhh1Δ*, and *pbp1Δ dhh1Δ* mutants growing in YPD and YPGL medium. mRNA levels were quantified by qRT-PCR analysis, and the relative mRNA levels were calculated using $2^{-\Delta\Delta Ct}$ method normalized to *SCR1* reference gene. The data show mean ± SEM (n = 3) of fold change of mRNA level from wild-type cells at 4 h of culture in YPGL. ns (not significant), $^{*}P < 0.05$, $^{**}P < 0.01$ as determined by Tukey's test.

## Pbp1 regulates mRNAs encoding mitochondrial proteins by inhibiting decapping factors and this regulation of Pbp1 requires both the coding sequence and 3'UTR of the mRNAs

The participation of Pbp1 in promoting proper deadenylation is in close interaction with Pab1 [18–20]. Though Pbp1 exhibited polysome distribution similar to that of Pab1, the association of Pbp1 with polysome was unaffected by the lack of Pab1 [18]. The *Drosophila* homolog of Pbp1, ATX2, influences translation independently from Pab1 homolog PABP [43]. These results raise the question of whether Pbp1 has specific functions in addition to generally affecting mRNA turnover through deadenylation. Our previous result proved that Pbp1 controls the translation of *HO* transcript by making a complex with Mkt1 [15]. We also showed that

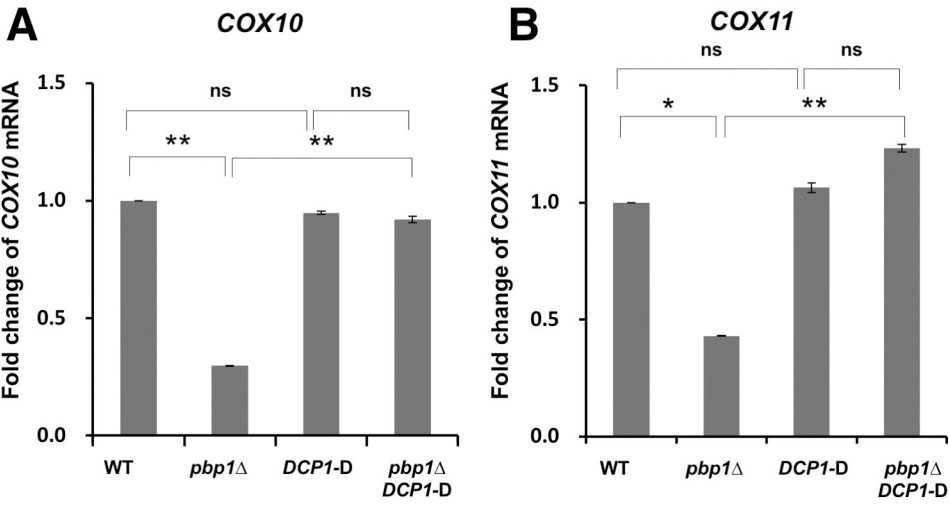

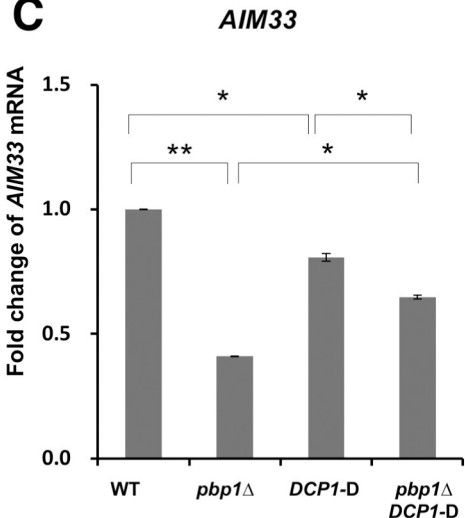

**Fig 14. Expression of *COX10*, *COX11*, and *AIM33* mRNA in wild-type, *pbp1Δ*, *DCP1-D*, and *pbp1ΔDCP1-D* strains growing in YPGL medium one hour after the addition of NAA.** The mRNA levels of (A) *COX10*, (B) *COX11*, and (C) *AIM33* in wild-type strain (WT), *pbp1Δ* mutant, the strain in which Dcp1 protein was fused with AID degron (*DCP1-D*), and *pbp1Δ* mutant with the same system (*pbp1Δ DCP1-D)* growing in YPGL medium one hour after the addition of NAA. mRNA levels were quantified by qRT-PCR analysis, and the relative mRNA levels were calculated using $2^{-\Delta\Delta Ct}$ method normalized to *SCR1* reference gene. The data show mean ± SEM (n = 3) of fold change of mRNA level from wild-type cells at 4 h of culture in YPGL. ns (not significant), $^*P < 0.05$, $^{**}P < 0.01$ as determined by Tukey's test.

Pbp1 enhanced the translation of *LRG1* mRNA in the absence of Pan2 [44]. Thus, these findings are examples in which Pbp1 regulates individual mRNA targets. In this study, Pbp1 regulates specific genes coding for proteins required for mitochondrial functions (*COX10*, *COX11*, *AIM3*, etc) in YPGL medium. Meanwhile, the expression levels of other mitochondrial genes including *MLS1*, *COX8*, *AIM7*, and *MCX1*, were not altered by the absence of Pbp1 (Table 3, S1A, S1B, S1F and S1G Fig). Therefore, our data support that the regulation of Pbp1 is not a global effect but rather selective.

Our results suggest that Pbp1 does not control the mRNA level through inhibiting the Pan2/Pan3 or Ccr4/Not complex, which process the deadenylation at the 3' UTR of mRNAs.

Rather, the effect of Pbp1 is related to decapping factors and 5'-3' exonuclease, including Dhh1, Dcp1, and Xrn1, which act at the 5' UTR of transcripts. In this model, Pbp1 prevents decapping and/or the 5'-to-3'-degradation of its bound mRNA, thereby stabilizing the transcripts. Another discovery in this study is that the regulatory effects of Pbp1 on *COX10* and *COX11* mRNAs require not only 3' UTR but also the coding sequence. These results suggest the model of an RNP close-loop in which Pbp1 interacts with decapping factors at the 5' UTR of its bound mRNAs and inhibits their activities. In fact, Pbp1 was reported to physically interact with Dhh1 [45]. In another study, loss of Dhh1 further enhanced the growth defect caused by the over-expression of Pbp1, suggesting an antagonizing relationship between Pbp1 and Dhh1 [22]. The *pbp1Δ* mutant also has a synthetic growth defect with the *dhh1Δ* mutant [46]. It has been showed that Pbp1 human homolog Ataxin-2 binds directly to particular mRNAs [47]. Therefore, Pbp1 may directly bind to the coding sequence and 3' UTR of particular mRNAs to antagonize Dhh1.

## Pbp1 and Ataxin2 may have similar functions

Our results suggest that Pbp1 regulates the expression of the genes involved in gluconeogenesis and mitochondrial function through different mechanisms: Pbp1 regulates the expression of the gluconeogenesis genes via their promoters; Pbp1 regulates the expression of some mitochondrial functional genes probably at the step of decapping through the coding sequence and 3'UTR of mRNA. The human ortholog of Pbp1, Ataxin-2, has been linked to many neurodegenerative diseases: mutations within Ataxin-2 such as poly(Q) expansion has been linked to the pathology of spinocerebellar ataxia type 2 [48]. Both Pbp1 and Ataxin-2 have conserved protein structures including the RNA-binding domains Like-Sm (LSm) and LSm-associated domain (LSm-AD) and the PAM domains for Pab1-binding and polyglutamine tract (poly (Q)), implying analogous functions of the two proteins. Similar to Pbp1, Ataxin-2 and its interactors are involved in post-transcriptional regulation. Individual target mRNAs of Ataxin-2 have been linked with neural physiology and homeostasis [49, 50]. In conclusion, our results facilitate a better comprehension of the role of Ataxin-2 protein in humans, which can lead to potential treatments for ATXN2-related neurological disorders.

## Supporting information

**S1 Table. Strains used in this study.**
(DOCX)

**S2 Table. Plasmids used in this study.**
(DOCX)

**S3 Table. Primers used in this study.**
(DOCX)

**S4 Table. Primers for RT-PCR used in this study.**
(DOCX)

**S1 Fig. Expression of *MLS1*, *COX8* and genes in group 3, 4, 5, 6, 7 of *pbp1Δ* mutant strain and wild-type strain growing in YPD and YPGL media.** mRNA expression of genes in (A) *MLS1*, (B) *COX8*, (C) genes in group 3 (*MRPL4*, *MRPL3*, *MRPL17*, *MRPL39*), (D) genes in group 4 (*MRPS35*, *MRP13*, *RSM25*, *RSM27*), (E) genes in group 5 (*MSY1*, *DIA4*, *SLM5*, *ISM1*, *MSF1*, *MSW1*) (F) genes in group 6 (*AIM33*, *AIM36*, *AIM11*, *AIM7*) (G) genes in group 7 (*IBA57*, *FMC1*, *AIM11*, *AIM7*) in wild-type strain (WT) and *pbp1Δ* mutants growing in YPD and YPGL media. mRNA levels were quantified by qRT-PCR analysis, and the relative mRNA

levels were calculated using $2^{-\Delta\Delta Ct}$ method normalized to *ACT1* reference gene. The data show mean ± SEM (n = 3) of fold change of mRNA level from wild-type cells at 4 h of culture in YPD. ns (not significant), $^{*}P < 0.05$, $^{**}P < 0.01$ as determined by Tukey's test.
(PPTX)

**S2 Fig. Expression of *GFP* reporter driven by promoters of *MRPL3*, *MRPS35*, *MSY1*, *AIM33*, and *IBA57* of *pbp1Δ* mutant strain and wild-type strain growing in YPD and YPGL media.** All the right panels display the mRNA levels of *GFP* gene driven by the promoter of (A) *MRPL3*, (B) *MRPS35*, (C) *MSY1*, (D) *AIM33*, and (E) *IBA57* in wild-type strain (WT) and *pbp1Δ* mutant strain growing in YPD and YPGL media. The left panels display the mRNA level of the corresponding endogenous genes in the same strain as the right panels. After extracted, the total mRNA is used to measure mRNA level of both endogenous genes and *GFP* gene. mRNA levels were quantified by qRT-PCR analysis, and the relative mRNA levels were calculated using $2^{-\Delta\Delta Ct}$ method normalized to *ACT1* reference gene. The data show mean ± SEM (n = 3) of fold change of mRNA level from wild-type cells at 4 h of culture in YPD. ns (not significant), $^{*}P < 0.05$, $^{**}P < 0.01$ as determined by Tukey's test.
(PPTX)

**S3 Fig. Level of *GFP* mRNA driven by *COX11* promoter with *ADH1* terminator and with *COX11* 3'UTR of *pbp1Δ* mutant strain and wild-type strain (WT) growing in YPD and YPGL media.** The level of *GFP* mRNA driven by *COX11* promoter with *ADH1* terminator (left panel) and with *COX11* 3'UTR (right panel). mRNA levels were quantified by qRT-PCR analysis, and the relative mRNA levels were calculated using $2^{-\Delta\Delta Ct}$ method normalized to *ACT1* reference gene. The data show mean ± SEM (n = 3) of fold change of mRNA level from wild-type cells at 4 h of culture in YPD. ns (not significant), $^{*}P < 0.05$, $^{**}P < 0.01$ as determined by Tukey's test.
(PPTX)

**S4 Fig. Dcp1 protein is degraded about 1 hour after the addition of Auxin.** The strain in which Dcp1 protein was fused with AID degron (*DCP1-D*), and *pbp1Δ* mutant with the same system (*pbp1Δ DCP1-D)* growing in YPGL media one hour after the addition of NAA. Dcp1-AID protein level was examined by Western blotting.
(PPTX)

## Acknowledgments

We thank the members of Molecular Cell Biology Laboratory for valuable discussions.

## Author Contributions

**Conceptualization:** Dang Thi Tuong Vi, Kenji Irie.

**Data curation:** Dang Thi Tuong Vi, Shiori Fujii.

**Formal analysis:** Dang Thi Tuong Vi, Shiori Fujii.

**Funding acquisition:** Kenji Irie.

**Investigation:** Dang Thi Tuong Vi, Shiori Fujii, Arvin Lapiz Valderrama, Ayaka Ito, Eri Matsuura, Ayaka Nishihata, Kaoru Irie, Kenji Irie.

**Methodology:** Dang Thi Tuong Vi, Shiori Fujii, Ayaka Ito, Eri Matsuura, Kaoru Irie, Yasuyuki Suda, Kenji Irie.

**Project administration:** Kenji Irie.

**Resources:** Kenji Irie.

**Supervision:** Kenji Irie.

**Validation:** Dang Thi Tuong Vi, Shiori Fujii, Kenji Irie.

**Visualization:** Dang Thi Tuong Vi, Shiori Fujii, Kenji Irie.

**Writing – original draft:** Dang Thi Tuong Vi, Shiori Fujii, Arvin Lapiz Valderrama, Yasuyuki Suda, Tomoaki Mizuno, Kenji Irie.

**Writing – review & editing:** Dang Thi Tuong Vi, Arvin Lapiz Valderrama, Yasuyuki Suda, Tomoaki Mizuno, Kenji Irie.

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
