## [Decision Letter · Decision Letter 0]

18 Mar 2021

PONE-D-21-05834

Pbp1, the yeast ortholog of human Ataxin-2, functions in the cell growth on non-fermentable carbon sources

PLOS ONE

Dear Dr. Irie,

Thank you for submitting your manuscript to PLOS ONE. After careful consideration, we feel that it has merit but does not fully meet PLOS ONE’s publication criteria as it currently stands. Therefore, we invite you to submit a revised version of the manuscript that addresses the points raised during the review process.

As you can see, the reviews are split on your manuscript.

While Reviewer 2 provided a positive response, Reviewer 1 raises a number of issues that should be addressed, especially regarding the design of experiments, and the conclusion of the manuscript without concrete evidence supporting the author’s claims.

After considering all of the points raised by the reviewers, and having looked carefully at the manuscript itself, I have come to the conclusion that we can reconsider your manuscript if you could address all the points raised by reviewers.

We look forward to receiving your revised manuscript.

Kind regards,

Reiko Sugiura, M.D., PhD.

Academic Editor

PLOS ONE

Journal Requirements:

2. Please include your tables as part of your main manuscript and remove the individual files. Please note that supplementary tables (should remain/ be uploaded) as separate "supporting information" files

Reviewers' comments:

Reviewer's Responses to Questions

**Comments to the Author**

1. Is the manuscript technically sound, and do the data support the conclusions?

Reviewer #1: Partly

Reviewer #2: Yes

2. Has the statistical analysis been performed appropriately and rigorously? 

Reviewer #1: I Don't Know

Reviewer #2: Yes

3. Have the authors made all data underlying the findings in their manuscript fully available?

Reviewer #1: Yes

Reviewer #2: Yes

4. Is the manuscript presented in an intelligible fashion and written in standard English?

Reviewer #1: Yes

Reviewer #2: Yes

5. Review Comments to the Author

Reviewer #1: The manuscript “Pbp1, the yeast ortholog of human Ataxin-2, functions in the cell growth on nonfermentable carbon sources” by Dang Thi Tuong Vi and colleagues investigated the effect of yeast knockout of PBP1, an orthologue of human Ataxin-2, on proliferation and transcriptional alterations when cells were grown in a medium containing glycerol and lactate.

The experiments and the conclusion of the paper are based on the observations in Figure 1, where the authors claim that growth inhibition was observed on non-fermentable carbon sources, but no quantitative experimental results are shown. Also, configurations of some media are missing. Without data on time-dependent growth of yeast strains on liquid media containing differential carbon sources, it is puzzling how they designed the experimental conditions for microarray and qPCR analysis.

The authors prepared several yeast strains themselves, but did not mention how they determined that the strains were properly constructed as intended. Even if the strains were properly constructed, the effect of restoring gene expression should be tested to discuss the effects of knockout or gene deletion, especially for knockout strains.

From the above perspective, in my opinion, the manuscript contains issues in design of experiments, and the conclusions of the manuscript are not supported by the results they presented. Therefore, I recommend to rejecting the manuscript.

Major points:

1. Fig.1 and Fig.5, Time-dependent growth at different carbon sources should be shown, as shown in Figures 1D and 1F of Reference 26 (Yang et al. 2019). Please clearly indicate the number of repetitions. If for any reason a spot assay needs to be performed, serial dilutions of the medium containing yeast cells should be used, and the number of repeats of the experiments should be clearly indicated as well.

2. Lines 122-132, Configurations of the solid medium (YPD, YPRaff, YPGL) used in Fig. 1 are missing.

3. For experiments using pbp1Δ, it is necessary to test the effect of restoring PBP1 expression.

Minor points:

4. Lines 130-131, describe the configurations of SC media.

5. Line 142, “10 mL”

6. For reagents and equipment used, indicate where the company is located (see recent research articles published in PLOS ONE, e.g. Hansen et al. doi: 10.1371/journal.pone.0247375).

7. Lines 165 and 179, Please cite Addgene plasmid number. (see https://help.addgene.org/hc/en-us/articles/205432559-How-do-I-cite-a-plasmid-that-I-received-from-Addgene-in-future-publications-)

Reviewer #2: The reviewer finds that the main finding of this paper is intriguing and the presented data are satisfactory. To make the conclusion of the paper more acceptable, the following minor points should be addressed.

p.20, l.330-331

Based on the observation that pbp1D and cat8D show no synthetic defect in the induction of PCK1 or FBP1 (Fig.4), it was concluded that the regulation of Pbp1 may not be mediated by the transcriptional activator Cat8. Generally speaking, a lack of synthetic genetic defects is regarded as indicative that both factors act in the same pathway, so the discussion here is somewhat confusing. As a synthetic growth defect was observed indeed (Fig.5), the reviewer agrees that it is more likely that Pbp1 and Cat8 act in different pathways. However, the interpretation of the data shown Fig.4 should be handled more carefully.

p.25, l.460-463

According to the Introduction section, glucose depletion induces TORC1 inactivation by Pbp1 through Pbp1 phosphorylation by Psk1 or methionine-rich region of Pbp1 (Pbp1 acts upstream to TORC1). On the other hand, discussion here sounds to presuppose a role of Pbp1 much closer to transcription (Pbp1 acts downstream of TORC1). Partly because the relationship between these two possibilities is not clear, the discussion here is confusing and should be revised.

Minor points (typo):

p.19, l.317

suppress catabolite

p.26, l.491

transcripts level

6. PLOS authors have the option to publish the peer review history of their article (what does this mean?). If published, this will include your full peer review and any attached files.

Reviewer #1: No

Reviewer #2: No

---

## [Author Response · Author response to Decision Letter 0]

21 Apr 2021

Our responses to the reviewers’ comments and changes in the revised manuscript

PONE-D-21-05834

Tittle: Pbp1, the yeast ortholog of human Ataxin-2, functions in the cell growth on non-fermentable carbon sources 

Authors: Dang Thi Tuong Vi, Shiori Fujii, Arvin Lapiz Valderrama, Ayaka Ito, Eri Matsuura, Ayaka Nishihata, Kaoru Irie, Yasuyuki Suda, Tomoaki Mizuno, Kenji Irie

Reviewer 1

1. Fig.1 and Fig.5, Time-dependent growth at different carbon sources should be shown, as shown in Figures 1D and 1F of Reference 26 (Yang et al. 2019). Please clearly indicate the number of repetitions. If for any reason a spot assay needs to be performed, serial dilutions of the medium containing yeast cells should be used, and the number of repeats of the experiments should be clearly indicated as well.

According to the reviewer’s comment, we have cultured WT strain and pbp1∆ strain in YPD, YPRaff, and YPGL liquid media and generated a time series data of Optical Density (OD A-600) with 3 replicates. We showed this data in the revised manuscript (Fig. 1D). The growth curve of pbp1∆ strain (doubling time 2.39 ± 0.08) is identical to that of WT strain (doubling time 2.57 ± 0.19) YPD liquid medium. Meanwhile, in YPGL liquid medium, the growth rate of pbp1∆ strain (doubling time 8.25 ± 0.18) is remarkably lower than that of WT strain (3.87 ± 0.05). The result of growth curve in liquid media is consistent with tetrad assay (Fig. 1A) and spot assay (Fig. 1B) which was conducted in 3 replicates. We also conducted the complementation assay in which we reintroduced PBP1 gene into pbp1∆ strain. In this experiment, the reintroduction of PBP1 gene recovered the growth deficiency of pbp1∆ strain in selective synthetic medium containing glycerol lactate (-UraGL) (Fig. 1C). We added these figures and edited the sentences in the revised manuscript (page 10, lines 231-233). We also generated a time series data for WT, pbp1∆, cat8∆, pbp1∆cat8∆ as Fig.6B and added text in the revised manuscript (page 15, lines 362-364)

2. Lines 122-132, Configurations of the solid medium (YPD, YPRaff, YPGL) used in Fig. 1 are missing.

We added the information about solid medium configurations in the revised manuscript (page 6, lines 130-137)

3. For experiments using pbp1Δ, it is necessary to test the effect of restoring PBP1 expression.

We conducted the complementation assay in which we reintroduced PBP1 gene into pbp1∆ strain. We also introduced into WT and pbp1∆ the backbone plasmid (YCplac33) to obtain control strains. We re-examined the expression of PCK1, FBP1, ICL1, COX10, COX11, CYT2, and PBP1 in theses strains. Unlike in YPGL medium, the difference expression level of PCK1, FBP, and ICL1 between wild-type and pbp1∆ was not observed in synthetic medium (-UraGL). The decreased expressions of COX10, COX11, CYT2, and PBP1 in the pbp1∆ mutant in -UraGL medium were efficiently rescued by the PBP1-containing plasmid. We added this data as shown in Fig.3 and explained the result in the revised manuscript (page 12-13, lines 294-306).

4. Lines 130-131, describe the configurations of SC media.

We added the configurations of SC media on page 6, lines 130-137.

5. Line 142, “10 mL”

We corrected this in the revised manuscript (page 7, line 149)

6. For reagents and equipment used, indicate where the company is located (see recent research articles published in PLOS ONE, e.g. Hansen et al. doi:10.1371/journal.pone.0247375).

We added this information in the revised manuscript (Materials and Methods section)

7. Lines 165 and 179, Please cite Addgene plasmid number. (see https://help.addgene.org/hc/en-us/articles/205432559-How-do-I-cite-a-plasmid-that-I-received-from-Addgene-in-future-publications-)

We cited Addgene plasmid numbers in revised manuscript (page 8, lines 174 and 187).

Reviewer 2

1. p.20, l.330-331

Based on the observation that pbp1∆ and cat8∆ show no synthetic defect in the induction of PCK1 or FBP1 (Fig.4), it was concluded that the regulation of Pbp1 may not be mediated by the transcriptional activator Cat8. Generally speaking, a lack of synthetic genetic defects is regarded as indicative that both factors act in the same pathway, so the discussion here is somewhat confusing. As a synthetic growth defect was observed indeed (Fig.5), the reviewer agrees that it is more likely that Pbp1 and Cat8 act in different pathways. However, the interpretation of the data shown Fig.4 should be handled more carefully.

In response to the reviewer’s comment, we re-interpreted the data with prudence in the revised manuscript (page 15, lines 364-366).

2.p.25,l.460-463

According to the Introduction section, glucose depletion induces TORC1 inactivation by Pbp1 through Pbp1 phosphorylation by Psk1 or methionine-rich region of Pbp1 (Pbp1 acts upstream to TORC1). On the other hand, discussion here sounds to presuppose a role of Pbp1 much closer to transcription (Pbp1 acts downstream of TORC1). Partly because the relationship between these two possibilities is not clear, the discussion here is confusing and should be revised.

We have modified the discussion to make the conclusion more explicit that Pbp1 is likely to act upstream of TORC1 (page 20, lines 482-488).

3.p.19,l.317

suppress catabolite

We modified this in the revised manuscript (page 14, lines 342)

4.p.26,l.491

transcripts level

We modified this in the revised manuscript (page 21, lines 515)

---

## [Decision Letter · Decision Letter 1]

27 Apr 2021

Pbp1, the yeast ortholog of human Ataxin-2, functions in the cell growth on non-fermentable carbon sources

PONE-D-21-05834R1

Dear Dr. Irie,

We’re pleased to inform you that your manuscript has been judged scientifically suitable for publication and will be formally accepted for publication once it meets all outstanding technical requirements.

Kind regards,

Reiko Sugiura, M.D., PhD.

Academic Editor

PLOS ONE

Additional Editor Comments (optional):

Reviewers' comments:

Reviewer's Responses to Questions

**Comments to the Author**

1. If the authors have adequately addressed your comments raised in a previous round of review and you feel that this manuscript is now acceptable for publication, you may indicate that here to bypass the “Comments to the Author” section, enter your conflict of interest statement in the “Confidential to Editor” section, and submit your "Accept" recommendation.

Reviewer #1: All comments have been addressed

Reviewer #2: All comments have been addressed

2. Is the manuscript technically sound, and do the data support the conclusions?

Reviewer #1: Yes

Reviewer #2: Yes

3. Has the statistical analysis been performed appropriately and rigorously? 

Reviewer #1: Yes

Reviewer #2: Yes

4. Have the authors made all data underlying the findings in their manuscript fully available?

Reviewer #1: Yes

Reviewer #2: Yes

5. Is the manuscript presented in an intelligible fashion and written in standard English?

Reviewer #1: Yes

Reviewer #2: Yes

6. Review Comments to the Author

Reviewer #1: (No Response)

Reviewer #2: (No Response)

7. PLOS authors have the option to publish the peer review history of their article (what does this mean?). If published, this will include your full peer review and any attached files.

Reviewer #1: No

Reviewer #2: No

---

## [Editor Report · Acceptance letter]

30 Apr 2021

PONE-D-21-05834R1 

Pbp1, the yeast ortholog of human Ataxin-2, functions in the cell growth on non-fermentable carbon sources 

Dear Dr. Irie:

I'm pleased to inform you that your manuscript has been deemed suitable for publication in PLOS ONE. Congratulations! Your manuscript is now with our production department. 

Kind regards, 

on behalf of

Dr. Reiko Sugiura 

Academic Editor

PLOS ONE